# The soil microorganism *Bacillus amyloliquefaciens* N3-8 shows potential as a biocontrol agent against the pathogen *Burkholderia pseudomallei* and its effect on rice plantation

Chotima Potisap,[1] Phrueksa Lawongsa,[2,3] Jittima Duangsri,[1,4] Júlia B. Gontijo,[5] Surasakdi Wongratanacheewin,[1,4] Jorge L. Mazza Rodrigues,[5] Rasana W. Sermswan[1,6]

**ABSTRACT** *Burkholderia pseudomallei* is a saprophytic bacterium responsible for melioidosis in humans and animals. In this study, *Bacillus amyloliquefaciens* N3-8 was applied as a biocontrol agent on sterile soil spiked with $10^7$ colony-forming unit (CFU) per gram of *B. pseudomallei* p37 at two ratios: 1:10,000 and 1:100,000 CFU/g soil. Both treatments significantly reduced *B. pseudomallei* by 4–5 logs within 4 weeks. A subsequent experiment applied the 1:10,000 ratio to 10 kg of natural soil in a pot containing $10^2$–$10^3$ CFU/g of *B. pseudomallei* alongside rice cultivation. Bacterial counts, rice yield, soil physicochemical factors, and microbial populations were monitored. *B. pseudomallei* was undetectable in biocontrol-treated soil by day 14 but reappeared by day 30, eventually matching the levels in control soil, suggesting interference by native microbial communities. No significant differences between the control and biocontrol treatments were observed in rice yield or soil physicochemical properties. Metataxonomic analysis revealed 17 bacterial phyla across all samples, with no significant differences in the overall microbial community structure between treatments at any time point. On the other hand, significant changes in microbial beta-diversity over time within the same soil treatments suggest that temporal dynamics, rather than the biocontrol treatment, drive shifts in microbial community structure. This study highlights the potential of *B. amyloliquefaciens* N3-8 as a biocontrol agent against *B. pseudomallei* on a pot scale with a rice plantation. For effective control of the pathogen, repeated applications in a rice field trial are necessary to ensure sustained management while being mindful not to disrupt the soil microbial balance.

**IMPORTANCE** *Bacillus amyloliquefaciens* N3-8 has been used in soil as a biocontrol agent against *Burkholderia pseudomallei*, a bacterium pathogenic to humans and animals, where it has shown no significant effects on soil physicochemical properties, rice yield, and bacterial community structure. However, long-term treatments are needed to achieve sustainable control, and critical management is required to avoid disturbing the microbial balance in the soil.

**KEYWORDS** biocontrol, melioidosis, rice plantation, soil

Melioidosis is an infectious disease found in both humans and animals (1). The disease has been observed predominantly in Southeast Asia and Northern Australia and was also reported in South Asia, such as China and India (2). Fatality rates of 30%–35% were reported in admitted patients in public hospitals in Thailand, whereas Australia reported 14% (3–5). Routes of infection are direct contact of an open wound with soil and water contaminated with *Burkholderia pseudomallei* or a lesser chance of inhalation and ingestion (5). A commercial vaccine is yet to be available, and disease

**Peer Reviewer** Lionel Moulin, IRD, PHIM Plant Health Institute, Montpellier, France

Address correspondence to Rasana W. Sermswan, rasana@kku.ac.th.

Chotima Potisap and Phrueksa Lawongsa contributed equally to this article. The author order was determined by those who were hands-on with the work and those who provided ideas and grant support.

The authors declare no conflict of interest.

treatment requires a long-term application of expensive antibiotics to prevent relapse, which is relatively high in melioidosis (6, 7). The human melioidosis cases worldwide were estimated to be 165,000, and around 89,000 died from this infection (2, 5). The disease is not zoonotic, and transmission from human to human is rare (8). Therefore, soil and water are the most important reservoirs of the bacterium, and controlling the pathogen in soil should reduce the risk of infection, especially for agricultural workers who are in frequent contact with soil and water when producing rice. *B. pseudomallei* is a free-living microorganism that was reported to survive in extreme environmental conditions (9). It can be isolated from various depths of the soil, 10 cm from the surface until 90 cm or more, depending on the physicochemical parameters of the soil and moisture content (10, 11).

Biological control is a bioeffector method for controlling one organism by another antagonistic organism through competition, parasitism, or antibiosis (12). For decades, many bacterial and fungal strains, viruses, nematodes, and insects have been used as biological control agents in managing soil-borne crop pathogens (13). Moreover, the prevalence of bacteriophages specific to human pathogens, such as *Escherichia coli* O157:H7, was reported and applied in food industries (14). However, through a literature search, none of this methodology has ever been used to control soil-borne human pathogens in the soil. *B. pseudomallei* is unevenly distributed in the soil of endemic areas as it was not detected in many agricultural soil samples, whether through cultivation on Ashdown's selective medium or molecular detection. Some soil physicochemical factors, such as pH and soil moisture, have been reported to be significantly different when comparing soils that are culture-positive or -negative for *B. pseudomallei* (11, 15). As a consequence, *Bacillus amyloliquefaciens* N3-8 was isolated from soils in Khon Kaen province, Thailand, which was found to be culture-negative for *B. pseudomallei* (16). Previously, we have shown that co-culturing of *B. amyloliquefaciens* N3-8 and *B. pseudomallei* in a liquid medium dramatically decreased the number of *B. pseudomallei* cells. We have also observed that secondary metabolites produced by *B. amyloliquefaciens* N3-8 inhibited the growth of *B. pseudomallei,* including drug-resistant isolates (17). *B. amyloliquefaciens* is commonly found in nature, and several strains have been reported to have probiotic potential, with some being used to produce enzymes in the food industries (18). Moreover, some organisms can produce various secondary metabolites and compete with other organisms living in the same environment (19, 20). We have yet, however, to test whether the application of a biocontrol agent, such as *B. amyloliquefaciens* N3-8, in agricultural soils can suppress the causing agent of melioidosis.

In this study, we investigated whether this soil microorganism has the potential to antagonize the human pathogen *B. pseudomallei*. First, we performed a series of co-culturing experiments of *B. pseudomallei* p37 with *B. amyloliquefaciens* N3-8 at different ratios to assess its ability to prevent the growth of *B. pseudomallei* p37. Second, we performed a greenhouse experiment in *B. pseudomallei*-containing soils, in which rice plants were grown to test the hypothesis that soil inoculation with *B. amyloliquefaciens* N3-8 can limit the number of this human pathogen. As no published data have demonstrated the use of a biocontrol strategy for controlling *B. pseudomallei* in the soil, we therefore demonstrated here, for the first time, using a non-pathogenic bacterium as a biocontrol agent for controlling a severe human pathogen in natural soil with rice plantation. The major physicochemical properties of soil and rice growing that served as a model plant were evaluated together with the bacterial diversity as observed by metataxonomics. This information is crucial to support the use of *B. amyloliquefaciens* N3-8 in the soil of endemic areas.

## RESULTS

### *Bacillus amyloliquefaciens* N3-8 is an effective biocontrol under laboratory conditions

Soil samples that tested positive for *B. pseudomallei* by direct culture were sterilized by autoclaving at 121°C for 15 min, spiked with a known amount of the bacterium, and re-cultured. The number of spiked *B. pseudomallei* in the sterile soil was $1 \times 10^7$ colony-forming unit (CFU)/g soil. When *B. amyloliquefaciens* N3-8 whole cells were used as a biocontrol agent, with the ratios of 1:10,000 or 1:100,000, only once or twice, the reduction in *B. pseudomallei* was observed since the first week, and it reduced the number of the pathogen by at least 5 log by the 4th week (Fig. 1).

### Biocontrol in the pot model with rice plantation

Natural soil with *B. pseudomallei* in plastic pots was planted with six rice seedlings as a plant model. There were four conditions of the rice pot: C1 was natural soil with *B. pseudomallei*, C2 was natural soil with *B. pseudomallei* and 1:10,000 CFU/g soil of *B. amyloliquefaciens* (biocontrol), F1 was C1 condition with fertilizer, and F2 was C2 condition with fertilizer (biocontrol). The number of *B. pseudomallei* at the start point in C1 was $1.4 \times 10^3$ CFU/g soil, and in F1 was $4.0 \times 10^2$ CFU/g soil. Soil samples were collected before rice seedlings were planted (day 0), during the tillering stage and before fertilizers were added (days 14 and 30), during the panicle formation stage (day 60), during the flowering stage (day 90), and before harvest (day 110). The decrease in the bacterium cell numbers in C2 and F2 of the biocontrol experiments could be observed since day 0 when the biocontrol agent was added and became uncultured on day 14 (Fig. 2). However, from day 30 onward, the number of *B. pseudomallei* in C2 and F2 could be detected but in a lower amount than their controls, except for F2 on day 30.

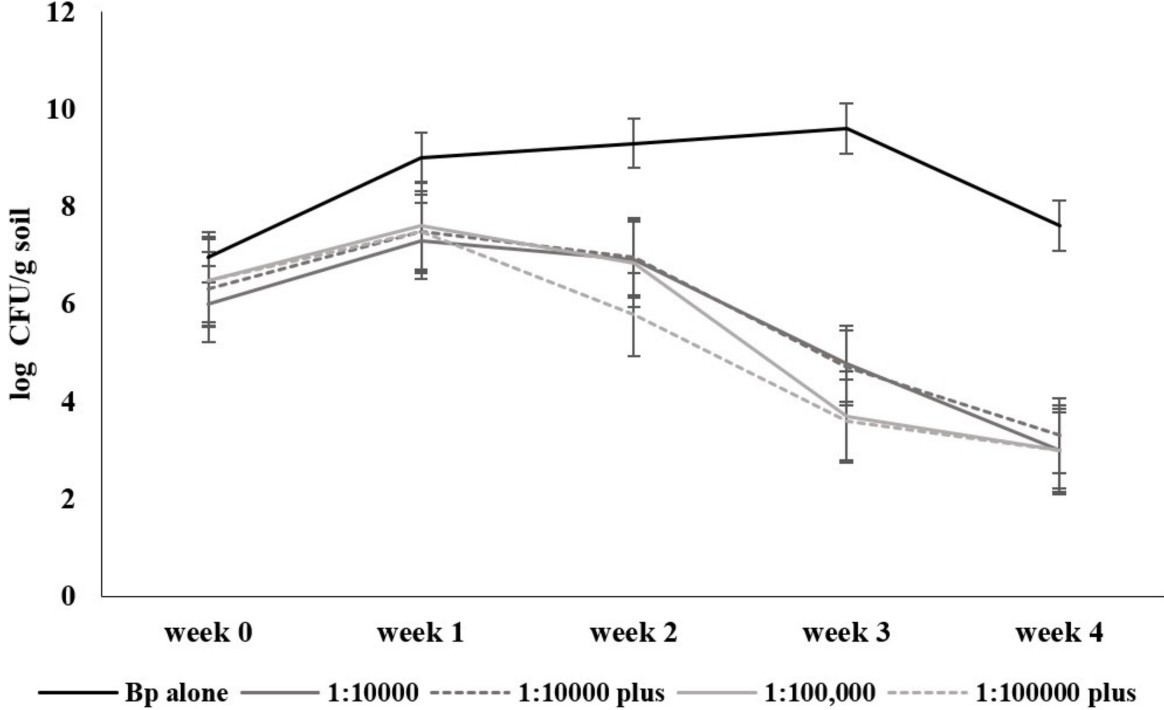

**FIG 1** The number of *B. pseudomallei* in the control and biocontrol soils in laboratory conditions. The experiments were done in five conditions: spiking soil with *B. pseudomallei* p37 control (black line), co-culture of *B. pseudomallei* p37:*B. amyloliquefaciens* N3-8 with CFU ratios of 1:10,000 (gray line) and 1:100,000 (light gray line), co-culture with CFU ratio of 1:10,000 with another addition (plus) at the 2nd week (dash gray line) and 1:100,000 with another addition (plus) at the 2nd week (dash light gray line). *B. pseudomallei* was cultured from soil of all conditions at 0, 1st, 2nd, 3rd, and 4th weeks. The *x*-axis is the time of investigation in weeks, and the *y*-axis is the number of *B. pseudomallei* in the soil in log CFU/g soil.

On days 90 and 110, the bacterium cell numbers increased in all conditions, and the amount in the biocontrol was not significantly different from their controls. There were no significant differences in the *B. pseudomallei* cell numbers when comparing C2 and F2, where the fertilizers used for growing rice were added.

### The effect of *B. amyloliquefaciens* N3-8 on the growth and yield of rice

To observe the effects of the biocontrol agent on the growth (as determined by height and weight) and yield of rice, we compared these parameters between C1 and C2 and F1 and F2. Statistical analysis showed no significant differences when comparing rice growing in control soil and biocontrol soil (C1 and C2) and between growing in control soil with fertilizer and biocontrol soil with fertilizer (F1 and F2) (Table 1). Our biocontrol agent, *B. amyloliquefaciens* N3-8 inoculation, did not increase any rice growth and yield parameters in the tested conditions.

### The effect of *B. amyloliquefaciens* N3-8 on soil physicochemical properties

The physicochemical properties of soil in the pots of all conditions and replicates were evaluated on days 0, 30, 60, 90, and 110 according to the rice stages as previously described. The physicochemical properties of biocontrol soils were not significantly different from those measured in control soils at each time point. Some parameters, such as $NH_4^+$ and $NO_3^-$, fluctuated at some time points in all conditions (Fig. 3).

### The effect of *B. amyloliquefaciens* N3-8 on the soil bacterial community

The soil in the pot model of all conditions and replicates collected on days 0, 14, 30, 60, 90, and 110 was used to analyze the diversity of the bacterial communities by amplification of the V4 region of the 16S rRNA gene and analyze the microbial metataxonomics (16S rRNA gene-based metagenomics analysis). All samples, except for F1 and F2 on day

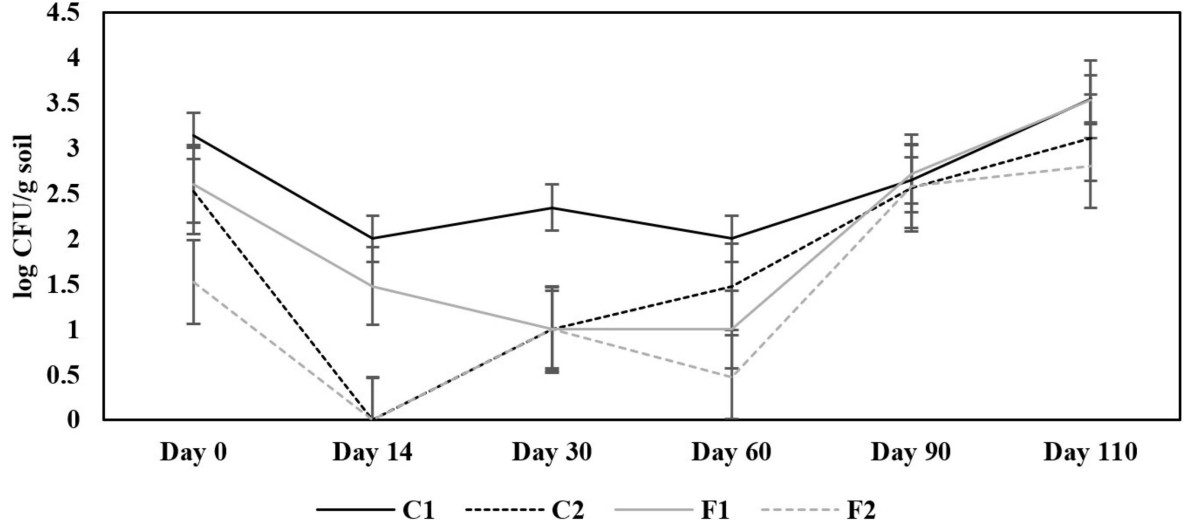

**FIG 2** The number of *B. pseudomallei* in the control and biocontrol soils of the pot model with rice plantation. *B. pseudomallei* culture-positive soil was adjusted to have 100% water holding capacity and put 10 kg each into 12 plastic pots. These pots were divided into four conditions of three replicates. (i) Soil in the pots planted with six rice seedlings was kept as a control (C1). (ii) Soil in the pots mixed with *B. amyloliquefaciens* N3-8 at the ratio of 1:10,000 CFU served as biocontrol (C2). (iii) Soil in the pots supplemented with fertilizer served as control with fertilizer (F1), and (iv) soil in the pots mixed with *B. amyloliquefaciens* N3-8 at the ratio of 1:10,000 CFU and supplemented with fertilizer served as biocontrol soil with fertilizer (F2). Six rice seedlings were planted in each pot as a model to observe the effect of biocontrol on plant growth and productivity. The soil samples were collected from each condition and replicated pot before rice planting (day 0), days 14 and 30 at the tillering stage, and fertilizers were added, day 60 at the rice panicle formation stage, day 90 at the rice flowering stage, and day 110 before harvest time. *B. pseudomallei* in each soil sample was cultured in Ashdown's selective medium and counted. The *x*-axis shows the days of soil collection, and the *y*-axis shows the numbers of *B. pseudomallei* in log CFU/g soil.

**TABLE 1** The growth and yield of rice in the pot model[a]

| Condition | Mean ± SD | | | | | |
|---|---|---|---|---|---|---|
| | Height of plant (cm) | Weight of plant (g) | Panicle number per plant | Weight of 100 grains (g) | Good grains/ Panicle | Bad grains/ Panicle |
| C1 | 69.87 ± 0.53 | 11.23 ± 0.44 | 3.33 ± 0.28 | 2.92 ± 0.23 | 45.67 ± 1.76 | 21.67 ± 1.45 |
| C2 | 67.23 ± 0.53 | 11.15 ± 0.44 | 3.83 ± 0.28 | 2.98 ± 0.23 | 50.33 ± 1.76 | 18.33 ± 1.45 |
| F1 | 70.85 ± 0.76 | 15.45 ± 0.76 | 5.98 ± 0.50 | 3.04 ± 0.04 | 72.10 ± 3.25 | 16.95 ± 2.51 |
| F2 | 80.00 ± 0.76 | 16.76 ± 1.57 | 6.18 ± 0.89 | 3.10 ± 0.41 | 82.68 ± 10.30 | 13.84 ± 1.74 |

[a]Wilcoxon signed-ranks test compared all parameters between C1 and C2 and F1 and F2. No significant difference was found.

0, yielded adequate reads and passed the quality parameters, making them suitable for further analysis.

The non-metric multidimensional scaling (NMDS) plot (Fig. 4) highlights the temporal dynamics of microbial community structure at the phylum level across treatments over time. The C1 (represented by circles) and C2 (represented by squares) show variability in their distribution within the NMDS space, indicating temporal changes. Microbial communities cluster together on days 0, 14, and 30, suggesting similar structures at the early stages. Beyond day 30, a spread in the NMDS space is observed, suggesting divergence in community structures over time. However, it is important to note that while the visual clustering in the NMDS plot might suggest differences between treatments, statistical analysis via permutational multivariate analysis of variance (PERMANOVA) did not detect significant differences in beta-diversity between the control (C1 and F1) and biocontrol (C2 and F2) treatments (Table 2; $P$-values: C1 vs C2 = 0.565, F1 vs F2 = 0.541, and overall = 0.245). The low $R^2$ values (0.0171 for C1 vs C2, 0.0256 for F1 vs F2, and 0.0384 overall) further indicate that the biocontrol treatment explains only a small fraction of the variance in beta-diversity. Contrastingly, significant temporal effects on beta-diversity were detected within the same treatments ($P = 0.001$ for each time comparison), emphasizing that changes in microbial community structure are primarily driven by time rather than the biocontrol treatment.

The microbial metataxonomics analysis revealed that among the 61 phyla of Bacteria and Archaea identified, 17 were consistently present with an average relative abundance above 0.1% (Fig. 5, Sequence Read Archive [SRA] under the accession number PRJNA1215386). All identified phyla belonged to Bacteria, with average relative abundances of all observed time points and conditions as follows: Acidobacteriota (13%), Actinobacteriota (6%), Armatimonadota (0.5%), Bacteroidota (10.5%), Bdellovibrionota (1.5%), Campylobacterota (0.1%), Chloroflexi (5.1%), Cyanobacteria (1.6%), Desulfobacterota (3%), Firmicutes (7.7%), Gemmatimonadota (1.9%), Myxococcota (6.7%), Nitrospirota (1.4%), Planctomycetota (4.1%), Proteobacteria (30.9%), Spirochaetota (0.5%), and Verrucomicrobiota (6.9%) (Fig. 5).

On day 0 in the C1 condition, the dominant phyla were Acidobacteriota (24.6%), followed by Proteobacteria (23.3%), and Chloroflexi (7.6%). Conversely, in the C2 condition on day 0, Proteobacteria was the most abundant at 25.4%, with Acidobacteriota at 24.9% and Myxococcota at 8.3%. Throughout the study, Proteobacteria emerged as the most prevalent phylum, with its abundance ranging from 20% to 50% across different time points. This phylum is the largest bacterial phylum and has a variety of roles in the ecosystem (21).

Additionally, the initial samples (day 0) exhibited higher proportions of Acidobacteriota and Proteobacteria, whereas samples from day 14 showed elevated levels of Proteobacteria and Firmicutes. However, these variations in phylum proportions were consistent across both control (C1 and F1) and biocontrol (C2 and F2) treatments, demonstrating no significant divergence between the two experimental setups (data not shown). This consistency underscores the uniformity in microbial community structure regardless of the treatment applied.

Analysis of the data by filtering amplicon sequence variant (ASV) counts at the genus level for *Burkholderia* and *Bacillus* illustrates that the relative abundance of *Bacillus*

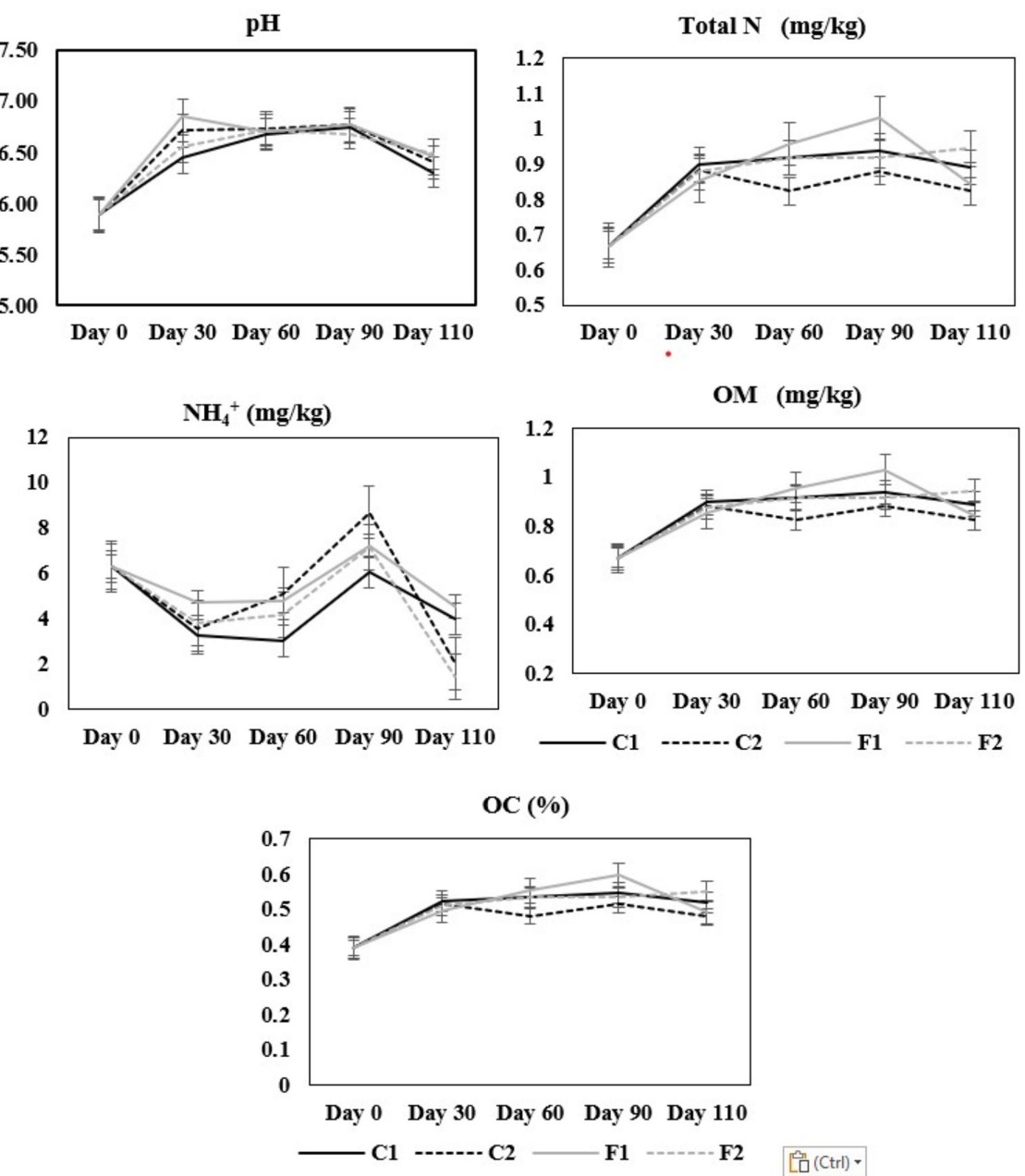

**FIG 3** Physicochemical properties of soil in the pot model. The conditions in the pot experiments were similar to what was described in Fig. 2. Soil samples were collected on days 0, 30, 60, 90, and 110 according to the stages of rice. The values of each parameter from the control soil (C1) were shown in the black line, the values from the biocontrol soil (C2) were shown in the dashed black line, the values in the control soil with fertilizer (F1) were shown in the gray line, and the values from the biocontrol soil with fertilizer (F2) were shown in the dashed gray line.

was significantly higher across all treatments compared to *Burkholderia*, which was undetectable on certain days (Fig. 6). Wilcoxon rank-sum test was used to compare the relative abundance of these genera across treatments, but no statistically significant enrichment or depletion was detected ($P = 0.468$ for C1 vs C2; $P = 1.000$ for F1 vs F2). A linear regression analysis conducted to assess potential correlations between *Bacillus* and *Burkholderia* across treatments showed no significant association.

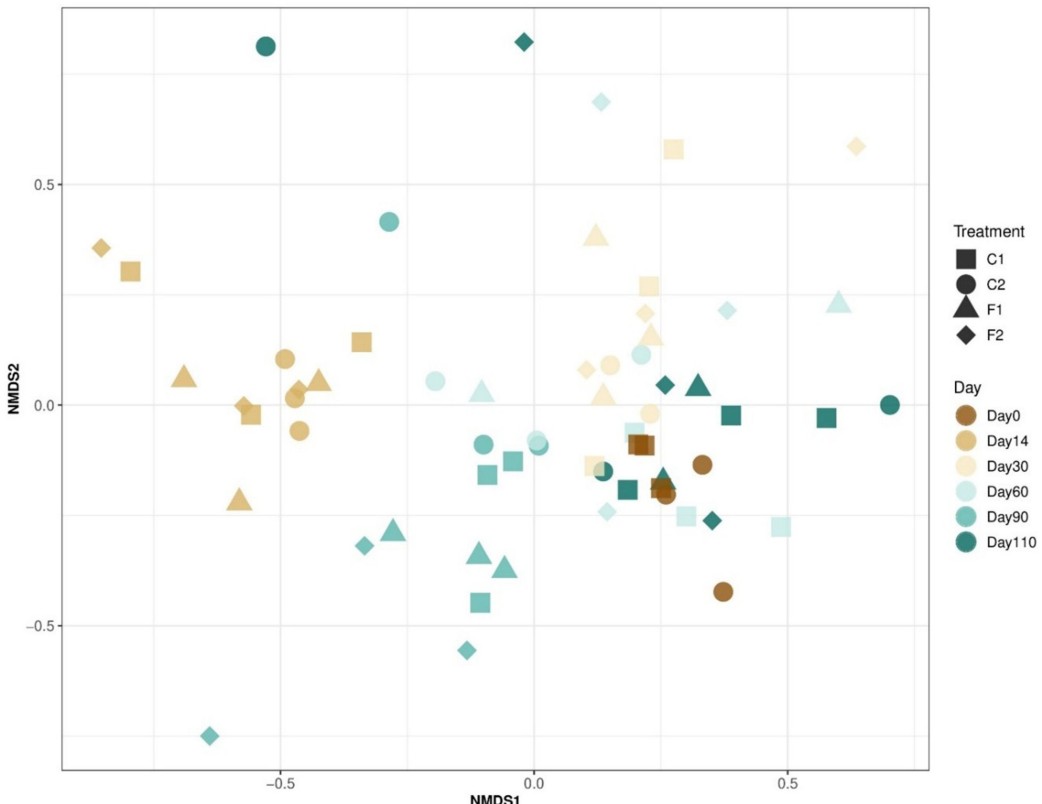

**FIG 4** NMDS plot of the beta-diversity of soil microbial community at the genus level. This scatter plot graph shows data points representing different conditions and days, with the axes labeled "NMDS1" and "NMDS2." The legend on the right categorizes the data points by treatment (C1, C2, F1, and F2) and by day (days 0, 14, 30, 60, 90, and 110).

## DISCUSSION

Soil and water are known to be the important reservoirs of *B. pseudomallei* that lead to infection in humans and animals when they come into contact with contaminated ones (22). The bacterium can survive through the dry season and nutrient depletion conditions, perhaps by biofilm protection, and then expand during the rainy season (9). A large number of patients infected with *B. pseudomallei* are rice agriculturists (23). Controlling the pathogens in soil should be beneficial to those who are at risk.

In a laboratory-scale experiment, a biocontrol treatment using *B. amyloliquefaciens* N3-8 of either 1:10,000 or 1:100,000 CFU ratio, with or without another addition of *B. amyloliquefaciens* N3-8, could significantly reduce a known amount of *B. pseudomallei* p37 spiked in sterile soil. However, when the 1:10,000 ratio was applied in the pot model with rice, it effectively decreased the number of this live pathogen to be unculturable within 14 days. Still, the pathogen became detectable in 30 days. Thereafter, this pathogen increased in both the control and the biocontrol soil. The amount

**TABLE 2** Permutational multivariate analysis of variance in beta-diversity of soil microbial community at the genus level

| Data | Days | | | Biocontrol | | | Days and biocontrol | | |
|---|---|---|---|---|---|---|---|---|---|
| | $R^2$ | $F$ | *P*-value | $R^2$ | $F$ | *P*-value | $R^2$ | $F$ | *P*-value |
| Overall | 0.3463 | 5.9910 | 0.001[a] | 0.0384 | 1.1197 | 0.245 | 0.1408 | 0.9370 | 0.676 |
| C1 vs C2 | 0.3965 | 3.8983 | 0.001[a] | 0.0171 | 0.8404 | 0.565 | 0.1184 | 1.1642 | 0.193 |
| F1 vs F2 | 0.3831 | 3.4111 | 0.001[a] | 0.0256 | 0.9119 | 0.541 | 0.0858 | 0.7640 | 0.902 |

[a]Significant statistical differences at *P*-value <0.05. Distance index: Bray-Curtis. $R^2$ (coefficient of determination), represents the proportion of the total variation in the data that is explained by a given factor. The *F*-value is a ratio used to test the significance of the factors. A higher *F*-value typically indicates a more significant factor.

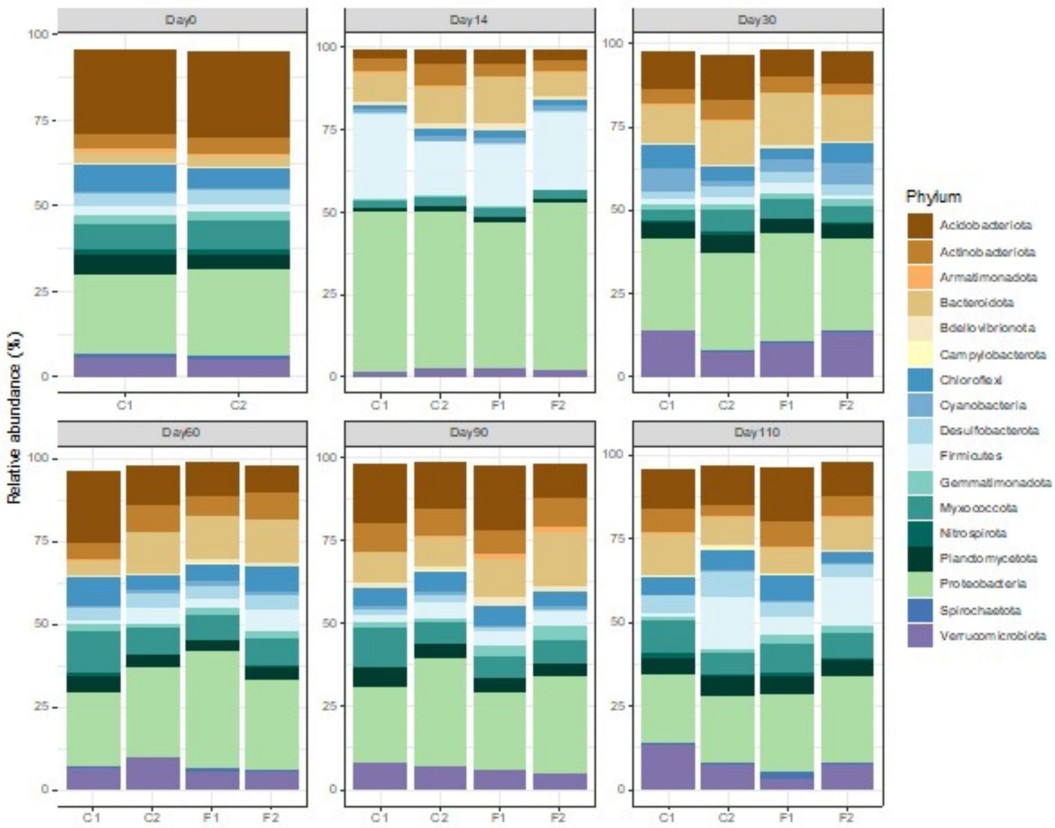

**FIG 5** The relative abundance of microbes in the control and biocontrol soils at the phylum level. The analysis was done using total DNA extracted from four soil conditions: C1 (control soil), C2 (biocontrol soil), F1 (control soil with fertilizer), and F2 (biocontrol soil with fertilizer) at six time points of 0, 14, 30, 60, 90, and 110 days.

of *B. pseudomallei* may fluctuate during the study but in a similar pattern between the control (C1) and fertilizer control (F1) conditions. Better control of *B. pseudomallei* observed on a small laboratory scale may partly be due to the close contact of *B. amyloliquefaciens* N3-8 secondary metabolites to kill the pathogen. An analysis of the composition of the secondary metabolites, for example, by mass spectrometry, may provide more clues to support our speculation. Moreover, a sterile soil condition could abolish the influences of other organisms in the soil. Effective control of the pathogen in natural soil conditions could mainly come from a combination of factors, including other microbial communities that may or may not support the activity of *B. amyloliquefaciens* N3-8. From our previous studies, we could isolate *B. amyloliquefaciens* isolates with killing activity against this pathogen from soils where *B. pseudomallei* was not detected (16, 17). Consistent application of the biocontrol or the identification of some soil physicochemical properties in natural soil that support *B. amyloliquefaciens* may help to sustain the biocontrol process. A report from Australia indicated the association of *B. pseudomallei* in the rhizosphere and the roots of specific grasses that may help the survival of the pathogen during the dry season (24). As rice is also a monocotyledon, this association could make the killing less effective and may require more than a one-time application of the biocontrol agent. A more in-depth observation of the association of the pathogen with the rice rhizosphere is needed, and it may help in designing more effective biocontrol strategies in the future.

Soil physicochemical parameters fluctuated at some time points across time and conditions in both control and biocontrol treatments but within the rice cultivation ranges, and did not show significant differences when compared between control and biocontrol conditions. Some *B. amyloliquefaciens* strains were reported to have

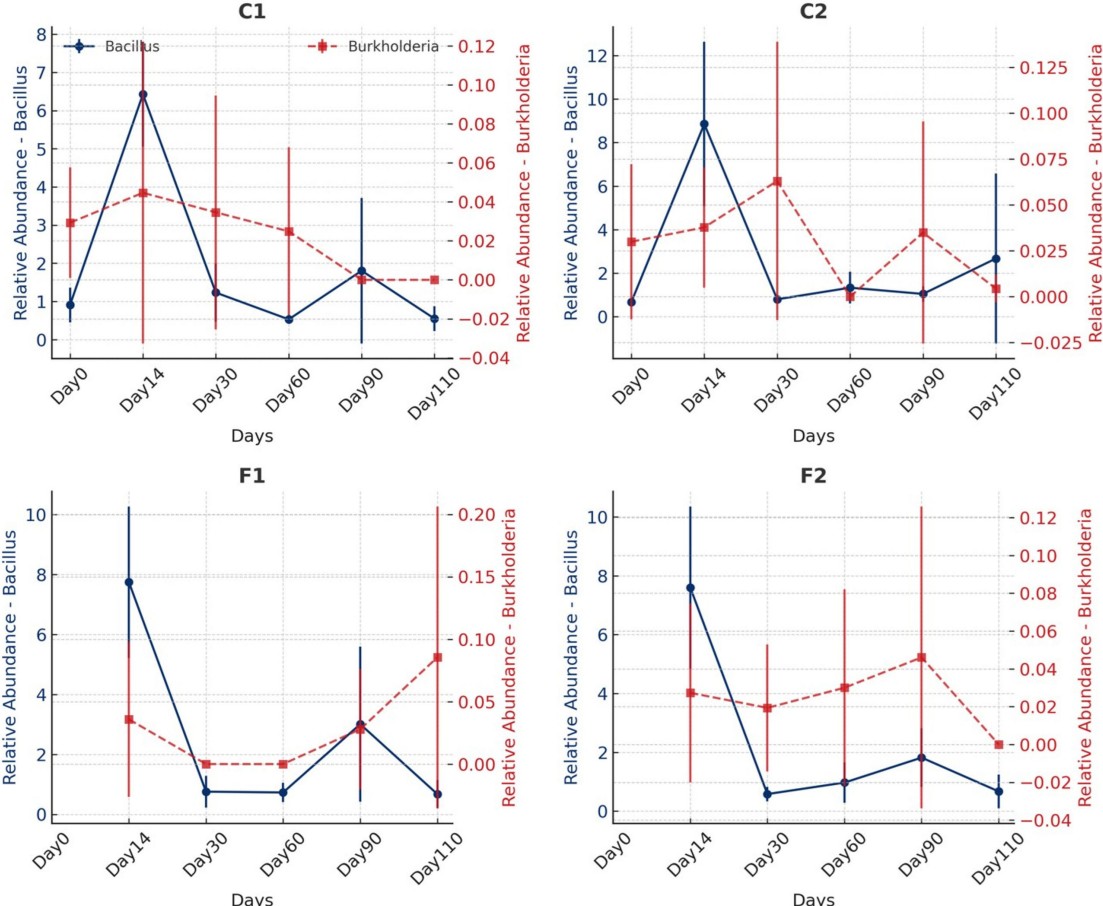

**FIG 6** The relative abundance of *Bacillus* and *Burkholderia* in the control and biocontrol soil. The analysis was done by filtering ASV counts at the genus level for *Burkholderia* and *Bacillus* in four soil conditions: C1 (control soil), C2 (biocontrol soil), F1 (control soil with fertilizer), and F2 (biocontrol soil with fertilizer) at six time points of 0, 14, 30, 60, 90, and 110 days.

growth-promoting activity for plants with different mechanisms. Analysis of chemical compounds in *B. amyloliquefaciens* FZB42 supernatant showed substances enhanced the production of indole-3-acetic acid, which is a plant hormone, that enhanced the growth of maize seeding (25). The *B. amyloliquefaciens* strain 54 was also reported to increase plant growth by enhancing the NPK (nitrogen-phosphorus-potassium) and chlorophyll content of plants (26). It was recognized as a potent biofertilization that can improve the N supply and, consequently, promote the growth of a wide spectrum of plants (27). However, we did not observe a significant difference in the growth or yield of rice between C1 and C2 or F1 and F2. A longer period of application and characterization of components in the secondary metabolites may be required to conclude an extra benefit of using *B. amyloliquefaciens* N3-8 in this case.

The application of fertilizers according to the normal process of rice plantation was required to obtain higher yield and growth, but the reduction of the pathogen in soil with fertilizer (C2 and F2) and without fertilizer (C1 and F1) supplementation was not significantly different. This was also supported by an experiment in which fertilizers were inoculated into the culture medium of *B. amyloliquefaciens* N3-8, and changes in survival and growth were not observed (data not shown). Therefore, biocontrol can be applied together with fertilizers, which is more convenient for rice growers.

Soil ecosystems comprise a complex array of interactions among diverse organisms such as bacteria, protozoa, nematodes, and fungi (28), contributing to the ecological balance (29). Our metataxonomic analysis has consistently detected 17 bacterial phyla

with a relative abundance above 0.1% across various conditions and time points, highlighting the rich microbial diversity present in the soil. Our results demonstrated that the microbial communities in both the C and F treatments were strongly influenced by the stages of rice. This indicates that temporal dynamics played a key role in shaping the microbial populations, suggesting that specific community structures may emerge at different stages of the sampling period. The assembly of microbial communities in rice is known to be closely linked to the plant's developmental stage, influencing the composition and interactions within the soil microbiome (30, 31). Furthermore, the biocontrol agent *B. amyloliquefaciens*, part of the phylum Firmicutes, has demonstrated efficacy in targeting the pathogen *B. pseudomallei* from the phylum Pseudomonadota. Crucially, the introduction of *B. amyloliquefaciens* does not significantly alter the overall microbial community structure, as evidenced by the stable proportions of Firmicutes in both control and biocontrol soils. This specificity is beneficial as it suggests that the biocontrol agent can suppress the target pathogen without disrupting the broader microbial ecosystem (32).

When analyzing the ASVs assigned as *Burkholderia* and *Bacillus,* we did not observe significant changes in the relative abundance of *Burkholderia* in response to *Bacillus* inoculation. Considering the conditions of our experiment, the *Bacillus* present in C1 and F1 were those naturally in the soil, while those in C2 and F2 were from natural plus *B. amyloliquefaciens* N3-8. A similar *Bacillus* pattern appeared in C1 and C2, and F1 and F2 support our finding. Furthermore, current sequence-based technologies are unable to differentiate between DNA from living and dead sources (33). Therefore, the relative abundance in C2 and F2 was obtained from both live and remnant DNA from the dead bacterium by the effects of the biocontrol agent. Notably, by day 14 post-inoculation, *B. pseudomallei* was no longer culturable on Ashdown's selective medium, indicating a potential shift in its viability rather than its relative abundance in sequencing data.

Given the uneven distribution of *B. pseudomallei* in the soil, it is advisable to optimize the application of *B. amyloliquefaciens* to areas with higher pathogen densities to enhance biocontrol efficacy while minimizing environmental impact. This targeted application strategy could prevent the overuse of the biocontrol agent and thereby reduce its ecological footprint. Moreover, the long-term dynamics of *B. amyloliquefaciens* in the soil, especially its ability to form spores and remain dormant, necessitate further study to understand its persistence and activity over time. This knowledge is crucial for determining the frequency and timing of biocontrol applications needed to maintain effective pathogen control.

Integrating biocontrol with other soil management practices such as organic amendments and crop rotations can also enhance the natural microbial suppression of pathogens (34). These practices support the biocontrol agent's function and improve overall soil health, creating a more resilient agricultural ecosystem (35). Lastly, the deployment of biocontrol agents must adhere to regulatory guidelines and consider community safety, especially in regions where *B. pseudomallei* is endemic. Gaining regulatory approval and community acceptance is crucial for the successful implementation of biocontrol strategies. Therefore, while *B. amyloliquefaciens* offers a promising approach to controlling *B. pseudomallei*, its application must be managed carefully to ensure ecological compatibility, effectiveness, and sustainability. This balance is vital for maintaining soil health and biodiversity while effectively controlling harmful human pathogens in endemic areas.

## Conclusion

The *B. amyloliquefaciens* N3-8, as a biocontrol agent, can be applied together with fertilizers to reduce the number of *B. pseudomallei* in soil without negative effects on the physicochemical properties of soil or beta-diversity of bacteria in the soil. The pot model here indicated that one-time treatment is not enough to effectively control this pathogen, which leads to the concern of management to minimize disturbance to the environment.

## MATERIALS AND METHODS

### Soil sampling

Twenty soil samples were randomly collected from a rice field (16°45'19.2"N 102°49'05.5"E) in the province of Khon Kaen, Thailand, at 15 cm depth. Individual soil samples were collected 1 m apart at the edges of the triangle and pooled together to obtain 1 kg. Soil samples were mixed in a sterile plastic bucket and sampled to cultivate *B. pseudomallei*. All positive soil sites were collected to be enough for all experiments, pooled, mixed, and brought back to our laboratory in sealed bags that were protected from sunlight. Negative soil sites were sterile by autoclaving at 121°C for 15 min and used in the biocontrol laboratory scale experiment.

### Bacterial strains

*B. amyloliquefaciens* N3-8 was isolated from soil in Khon Kaen province, Thailand, an area considered endemic for *B. pseudomallei* (17). The bacterium was cultured in Luria Bertani (LB) broth with 200 rpm shaking at 37°C for 24 h to obtain $1 \times 10^9$ CFU/mL. *B. pseudomallei* p37 was isolated from a sepsis patient admitted to Srinagarind Hospital, Khon Kaen, Thailand (36). The bacterium was cultured in LB at 37°C with agitation at 200 rpm.

### Detection of *B. pseudomallei* in soils

The direct culture method was used to investigate the presence of *B. pseudomallei* in 100 g of each soil sample from a rice field and the pot model (37). Suspected *B. pseudomallei* colonies grown on Ashdown agar that were wrinkled and appeared violet-purple were confirmed by the latex agglutination test (38).

### Physicochemical properties of soil

The texture of the initial soil was determined using the pipette method (39), which showed a loamy sand texture with 74.42% sand, 15.54% silt, and 10.04% clay. Interval soil samples of 300 g were randomly collected from the pot at the depth of the plow layer (0 cm–15 cm) before rice seedlings were planted on day 0, at the tillering stage on day 30, at the panicle formation stage on day 60, at the flowering stage on day 90, and day 110 before the rice was harvested. The soil samples were air-dried and then finely ground to pass through a 2 mm sieve. The soil pH was measured using a portable pH/mV/°C meter (HI8424, HANNA Instruments, Rhode Island, USA). The organic matter content and organic carbon content were determined by the Walkley and Black method (40), and the total nitrogen was measured using the micro-Kjeldalh method (41). Soil mineral N (ammonium, $NH_4^+$-N, and nitrate; $NO_3^-$-N) was extracted from 10 g fresh soil samples with 50 mL 2 M KCl and was analyzed by MgO-Devarda's alloy steam distillation as described by Keeney and Nelson (42).

### Biocontrol experiments

#### *Laboratory scale*

The growth curve of *B. pseudomallei* p37 was obtained by inoculating 1% of an overnight culture of *B. pseudomallei* p37 in LB broth and cultured at 37°C on a rotary shaker at 200 rpm. The growth was measured as the increase in optical density at 600 nm (OD600) every hour together with a spread plate method to count the colonies so that the OD600 of *B. pseudomallei* p37 culture can be converted to obtain CFU. The culture of *B. pseudomallei* p37 was thoroughly mixed with 2 kg of sterile soil, adjusted to 100% water holding capacity (WHC), to achieve a concentration of $1 \times 10^7$ CFU/g of soil. Ten grams of this mixed soil was used to culture *B. pseudomallei* to confirm the exact bacterial count. The mixed soil was divided into 75 tubes of 10 g each in sterile 50 mL self-standing plastic tubes with caps. These tubes were set for five conditions (15 tubes in each

condition) and tested in triplicate at each time point. They were used for direct culture of *B. pseudomallei* at 0, 1st, 2nd, 3rd, and 4th weeks. These five conditions consisted of *B. pseudomallei* p37 control, co-culture of *B. pseudomallei* p37:*B. amyloliquefaciens* N3-8 with CFU ratios of 1:10,000 and 1:100,000, and co-culture with CFU ratios of 1:10,000 and 1:100,000 with addition of *B. amyloliquefaciens* N3-8 with the same ratio at the 2nd week. All tubes were thoroughly mixed before being incubated at room temperature with loosely closed caps.

### *Pot scale*

The experimental design started from 12 plastic pots 30 cm in diameter. *B. pseudomallei* culture-positive soil was used in this study. The soil was thoroughly mixed and adjusted to 100% WHC. Each pot contained 10 kg of prepared soil, and the water level of each pot was kept 3 cm above the soil level. Four pot model conditions with three biological replicates consisted of *B. pseudomallei*-positive soil (control soil, C1), *B. pseudomallei* plus *B. amyloliquefaciens* N3-8 at the ratio of 1:10,000 CFU (biocontrol soil, C2), *B. pseudomallei*-positive soil supplement with fertilizer (control soil with fertilizer, F1), and *B. pseudomallei* plus *B. amyloliquefaciens* N3-8 at a ratio of 1:10,000 CFU and supplement with fertilizer (biocontrol soil with fertilized, F2). Rice was used as a model to observe the effect of biocontrol on plant growth and productivity. Six rice seedlings were grown in each pot. Chemical fertilizers with an NPK ratio of 15-15-15 (g/kg) were applied 14 days after planting, during the tillering stage, followed by the application of a 46-0-0 nitrogen fertilizer 30 days after planting, at the maximum tillering stage. The soil samples were collected on day 0, before rice seedlings were planted, days 14 and 30 at the tillering stage before fertilizers were applied, day 60 at the panicle formation stage, day 90 at the flowering stage, and day 110 before harvest from each condition and replicated pot for *B. pseudomallei* culture in Ashdown's selective medium and metataxonomics analysis. Soil collection on day 14 was used for bacterial culture and metataxonomics to observe the early changes of the microbes in the soil but not for the physicochemical properties of the soil.

### Rice growth and yield parameters

Rice growth was assessed by measuring plant height (cm) and weight (g), while rice yield was evaluated based on the number of panicles per plant, the weight of 100 grains (g), and the number of grains per panicle. The data of all parameters were compared between C1 and C2 and F1 and F2 by Wilcoxon signed-ranks test.

### Total DNA isolation and sequencing

Total DNA was extracted from each soil sample using the PowerSoil DNA isolation kit (MO-BIO Laboratories, Carlsbad, CA, USA) following the manufacturer's recommendations. The DNA concentration was required to be above 30 ng/µL, with an OD260/280 ratio between 1.8 and 2.0. DNA quality was further assessed by agarose gel electrophoresis, ensuring the absence of smear patterns. The extracted DNA was stored at −20°C until use.

The diversity of microbial communities was assessed by high-throughput sequencing of the V4 region of the 16S rRNA gene, using the 515F (43) and 806R (44) primer pair. PCR was performed in triplicate using the Phusion Hot Start II High-Fidelity PCR Master Mix (Thermo Scientific Inc., Waltham, MA). Each reaction contained the following: 10 µL of Master Mix, 1 µL of each primer (10 pMol), 1 µL of the DNA template (10 ng/µL), and 7 µL of $H_2O$. PCR reactions were conducted with the C1000 Touch Thermocycler (Bio-Rad Laboratories, Inc., Hercules, CA) using the following protocol: initial denaturation (98°C for 30 s), 30 cycles of denaturation (98°C for 15 s), annealing (50°C for 30 s), and elongation (72°C for 15 s), followed by a final extension (72°C for 10 min). Paired-end sequencing, with 250 bp reads, was performed using the Miseq platform (Illumina,

Inc.) at the University of California-Davis Genomics Facility, Davis, CA, according to the manufacturer's instructions.

## Bioinformatics and statistical analyses

All bioinformatics and statistical analyses were performed on RStudio 4.3.1 (45). Raw sequences were processed by inferring the ASVs using the Dada2 1.28.0 package (46). We obtained 4,378,031 sequencing reads. Forward and reverse reads with a Phred score <30 were truncated at position 245 and 180 base pairs, respectively, and processed for filtering, trimming, error correction, dereplication, merging, and chimera removal. After quality control, a total of 2,806,526 microbial sequences with an average length of 225 bp were obtained. Taxonomy was assigned using the SILVA reference database (release 138.1, 07.03.2021). The ASV counts classified in the same taxonomic groups were summarized and normalized to within-sample relative abundance following the total sum scaling method (47). Microbial taxa were grouped at the phylum and genus levels for further analysis.

The graphical visualization were performed using vegan 2.6-4 (48), R packages and ggplot2 3.4.3 packages performed on RStudio 4.3.1 (45). NMDS and PERMANOVA followed by False Discovery Rate (FDR) adjustment were used to estimate the similarities among different treatments (Bray-Curtis distance) at the genus level. Two-way analysis of variance of aligned rank transformed data was used to investigate the effect of each treatment and sampling day on the relative abundances of the most abundant phyla (>0.1%), then followed by pairwise comparisons using FDR adjustment. The ASVs assigned as *Burkholderia* and *Bacillus* were manually filtered, and a Mann-Whitney U-test was performed to compare the abundance of *Burkholderia* between treatments C1 vs C2 and F1 vs F2 ($P < 0.05$).

## ACKNOWLEDGMENTS

Dr. Chotima Potisap was supported by Post-doctoral training scholarship PD2564-08, Khon Kaen University. The National Research Council of Thailand (NRCT) (N35A650543) funds this project.

## AUTHOR AFFILIATIONS

[1]Melioidosis Research Center, Khon Kaen University, Nai Mueang, Khon Kaen, Thailand

[2]Department of Soil Science and Environment, Faculty of Agriculture, Khon Kaen University, Nai Mueang, Khon Kaen, Thailand

[3]Soil Organic Matter Management Research Group, Khon Kaen University, Nai Mueang, Khon Kaen, Thailand

[4]Department of Microbiology, Faculty of Medicine, Khon Kaen University, Nai Mueang, Khon Kaen, Thailand

[5]Department of Land Air and Water Resources, University of California Davis, Davis, California, USA

[6]Department of Biochemistry, Faculty of Medicine, Khon Kaen University, Nai Muaeng, Khon Kaen, Thailand

## AUTHOR ORCIDs

Júlia B. Gontijo http://orcid.org/0000-0003-1942-7242
Surasakdi Wongratanacheewin http://orcid.org/0000-0003-2921-2536
Jorge L. Mazza Rodrigues http://orcid.org/0000-0002-6446-6462
Rasana W. Sermswan http://orcid.org/0000-0002-3691-5064

## AUTHOR CONTRIBUTIONS

Chotima Potisap, Conceptualization, Data curation, Formal analysis, Investigation, Methodology, Validation, Writing – original draft | Phrueksa Lawongsa,

Conceptualization, Investigation, Methodology, Supervision, Validation, Writing – review and editing | Jittima Duangsri, Formal analysis, Methodology, Software, Validation, Writing – review and editing | Júlia B. Gontijo, Data curation, Formal analysis, Software, Writing – review and editing | Surasakdi Wongratanacheewin, Funding acquisition, Methodology, Validation, Writing – review and editing | Jorge L. Mazza Rodrigues, Conceptualization, Funding acquisition, Methodology, Supervision, Writing – review and editing | Rasana W. Sermswan, Conceptualization, Funding acquisition, Methodology, Project administration, Supervision, Visualization, Writing – review and editing

## DATA AVAILABILITY

The 16S rRNA sequencing data set generated and analyzed during the current study is available in NCBI's Sequence Read Archive (SRA) under the accession number PRJNA1215386.

## ADDITIONAL FILES

The following material is available online.

### Open Peer Review

**PEER REVIEW HISTORY (review-history.pdf).** An accounting of the reviewer comments and feedback.

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
