## [Reviewer comments · Microbiology Spectrum]

Microbiology Spectrum

The soil microorganism *Bacillus amyloliquefaciens* N3-8 shows potential as a biocontrol agent against the pathogen *Burkholderia pseudomallei* and its effect on rice plantation

Chotima Potisap, Phrueksa Lawongsa, Jittima Duangsri, Julia Gontijo, Surasakdi Wongratanacheewin, Jorge Rodrigues, and Rasana Sermswan

Corresponding Author(s): Rasana Sermswan, Faculty of Medicine, Khon Kaen University

Review Timeline:

Submission Date:	December 2, 2024
Editorial Decision:	January 22, 2025
Revision Received:	March 13, 2025
Accepted:	March 17, 2025

Editor: *Frédérique Reverchon*

Reviewer(s): Disclosure of reviewer identity is with reference to reviewer comments included in decision letter(s). The following individuals involved in review of your submission have agreed to reveal their identity: *Lionel Moulin (Reviewer #2)*

Transaction Report:

DOI: <https://doi.org/10.1128/spectrum.02963-24>

Re: Spectrum02963-24 (The soil microorganism *Bacillus amyloliquefaciens* N3-8 is a biocontrol agent for the pathogen *Burkholderia pseudomallei* in rice plantation)

Dear Dr. Rasana W. Sermswan:

Thank you for the privilege of reviewing your work. Below you will find my comments, instructions from the Spectrum editorial office, and the reviewer comments.

I have received comments made by two independent reviewers. They both highlight the merits of your study whilst providing comments to improve some aspects of your manuscript, such as focusing your discussion on statistically significant results.

A data availability statement should also be added to your manuscript.

Revision Guidelines

Sincerely,
Frédérique Reverchon
Editor
Microbiology Spectrum

Reviewer #1 (Public repository details (Required)):

The authors did not indicate in their manuscript if they deposited their metagenomic data in a repository.

Reviewer #1 (Comments for the Author):

The document is about the use of the soil microorganism *Bacillus amyloliquefaciens* N3-8 as a biocontrol agent for the pathogen *Burkholderia pseudomallei* in rice plantations to prevent melioidosis. *Burkholderia pseudomallei*, the causative agent of melioidosis, poses significant health risks, particularly in Southeast Asia and Northern Australia. The authors identified *Bacillus amyloliquefaciens* N3-8 as a potential biocontrol agent against *B. pseudomallei* and evaluated the effectiveness of *B. amyloliquefaciens* N3-8 in reducing *B. pseudomallei* proliferation in both controlled laboratory settings and more naturalistic pot experiments with rice plants. However, the paper needs thorough language editing.

Major comments

1. The authors should showcase the limitations of the study in the discussion and abstract as this will help justify the methods they employed in the study.
2. From the authors conclusions the biocontrol treatment did not significantly affect the rice yield; showed no significant differences when compared between the control soil and biocontrol soil (C1 and C2) and between the control soil with fertilizer and biocontrol soil with fertilizer (F1 and F2). Invariably the biocontrol agent had no plant growth-promoting properties. It would have provided some clarity if authors characterized the secondary metabolite secreted by the bacillus strain to ascertain it had no plant growth-promoting properties. As it is rare for *Bacillus* strain not to possess both biocontrol and PGP properties in tandem, the authors should provide past work/research to justify the conclusions that biocontrol agent could only antagonize *B. pseudomallei* and not promote plant growth simultaneously.
3. Figures: In general, the figure legends need clearer information and should be formatted per the journal requirements as well as up to the standard required for publication.

Other minor comments can be found in the attached PDF file

Reviewer #2 (Comments for the Author):

Review on " The soil microorganism *Bacillus amyloliquefaciens* N3-8 is a biocontrol agent for the pathogen *Burkholderia pseudomallei* in rice plantation" by Potisap et al. for *Microbiology Spectrum* Journal.

In this article the authors have evaluated the impact of co-inoculating a *Bacillus amyloliquefaciens* strain (shown previously to antagonize *Burkholderia pseudomallei* in laboratory) in a soil containing a defined concentration of *B. pseudomallei* cells. They monitored the number of *B. pseudomallei* at different times post inoculation, and monitored the impact during a rice cycle in a pot experiments, on *B. pseudomallei* counts, soil physico chemical structure, rice growth parameters, and bacterial community structure (using 16S V4 amplicon barcoding). The *Bacillus* inoculation reduced the CFU of *B. pseudomallei* in soil but its impact did not eradicate the pathogen which growth kinetics increased at later times. The *Bacillus* inoculation had no statistically supported impact on rice growth parameters, and the bacterial communities changed over time but authors did not look into details at genus level to see enrichment or depletion between conditions. Globally the study is interesting and the topic to use bacterial inoculant to biocontrol human pathogens in soils is interesting. I have made several major and minor comments to improve the manuscript.

Major comments:

1. The title of the manuscript is dedicated to biocontrol against *B. pseudomallei*, but half of the work is also dedicated to monitor the impact of *Bacillus* inoculation on rice productivity. The title and manuscript should better integrate the two dimensions of the potential of *Bacillus* inoculation: biocontrol and plant growth promoting properties on rice
2. The control used to compare *Bacillus* inoculation in soil against *B. pseudomallei* is soil without inoculation. A good control would have been a bacterial strain with no known PGPR or biocontrol effect, as an *E. coli* for example. Because the fact to inoculate a number of bacterial cells in exponential phase in high numbers can have an impact whatever their biocontrol capacities. It could have revealed the specificity and particularities of *Bacillus* compared to others.
3. The 16S amplicon barcoding approach could be used also to monitor *B. pseudomallei* counts, even if the V4 region of 16S is not highly specific, I guess the authors have looked at *B. pseudomallei* counts in their ASV count table. Did they identify the *B. pseudomallei* or *Burkholderia* genus? These data could be interesting to look at. On the other hand, given the authors have the DNA for metagenomics, they could use PCR or qPCR to identify and quantify *B. pseudomallei* in the rice pot experiment. There are qPCR primers described in the literature to quantify *B. pseudomallei* cells.
4. Authors should evaluate genus enrichments or depletion across conditions in their 16S amplicon barcoding data. If no main differences are observed (apart from time) globally it does not mean that specific genera (or even species) cannot be detected using specific tests (Wilcoxon tests for examples). It could be interesting at least to look at the *Burkholderia* and *Bacillus* genera

in particular, given these have been inoculated (or not) between conditions.

5. The part on the impact of inoculation on soil physicochemical parameters should be shortened and main results better underlined, and shortened in the discussion while focusing on main impact (which appears to be on nitrogen, but why authors did focus on nitrogen ? and not other nutrients?).

Minor comments

In all the manuscript, authors give the strain name of *Bacillus*, while they never give it for *B. pseudomallei* except in the material and methods. Authors should indicate in the text everywhere they mention *B. pseudomallei* that it is strain p37.

Line 48 to 52: rephrase to "*B. amyloliquefaciens* N3-8 has been used in soil as a biocontrol agent against *B. pseudomallei*, a bacterium pathogenic to humans and animals, where it has shown significant effects on soil physicochemical properties, rice yield and bacterial community structure. However, long-term treatments are needed to achieve sustainable control and critical management is required to avoid disturbing the microbial balance in the soil"

Line 107: "autoclaved sterile": change to : sterilized by autoclaving . Authors should indicate conditions of autoclaving (one or 2 times, which temperature and pressure), as soil autoclaving conditions can be critical in achieving good sterilization.

Line 158: "The effect of *B. amyloliquefaciens* N3-8 on growth and yield of rice". I don't understand if there is an effect of fertilization or not on rice yield? The table 1 does not indicate any significant impact of either fertilization or bioinoculation on rice yield. If this is the case it should be clearly written in the text that *Bacillus* inoculation did not increase any rice growth parameters in the tested conditions.

Line 188: The authors should be careful when describing their NMDS plot, as they did mention differences in the text but apart from a time impact on beta diversity, there is not statistical difference between treatments when they tested it in Table 2 .

Line 235: Figure 5: in this figure authors should include at least a binning plot at genus level to monitor *Burkholderias*. Showing diversity binning plots at order or class level is not informative at all. To detect differences between treatments, authors could have performed enrichment tests to see at genus level if there are enriched genera in the treatments, using Kruskal-Wallis or Wilcoxon tests. This is common tests in amplicon barcoding analyses.

Line 268-269 ; "Better control of *B. pseudomallei* observed on a small laboratory scale may be due to the presence of secondary metabolites produced by *B. amyloliquefaciens* N3-8, causing the killing of the pathogen": this is purely speculative, it could be the result of many different other things, if authors want to test this hypothesis they should make experiments to reveal secondary metabolites production in their pot condition, or at least give insights in the discussion on how to test it to prove it.

Line 357-361: direct detection of *B. pseudomallei*. Why no molecular methods were used to confirm the belonging of the strains to *B. pseudomallei* species? What is the robustness and specificity of the latex agglutination method? Specific primers do exist to identify *B. pseudomallei*.

Line 339-344 (conclusion): The results do not show a strong decrease of *B. pseudomallei* in the long term, authors should be more cautious on their conclusion to not oversell the impact of inoculation. There are no statistically significant impact of the inoculation on any rice growth parameters, and biocontrol of *B. pseudomallei* did not last.

The soil microorganism *Bacillus amyloliquefaciens* N3-8 is a biocontrol agent for the pathogen *Burkholderia pseudomallei* in rice plantation

Chotima Potisap^{a*}, Phrueksa Lawongsa^{b,c*}, Jittima Duangsri^{a,d}, Júlia B. Gontijo^e, Surasakdi Wongratanacheewin^{a,d}, Jorge L Mazza Rodrigues^e, Rasana W Sermswan^{a,f,#}

^a Melioidosis Research Center, Khon Kaen University, Khon Kaen, Thailand

^b Department of Soil Science and Environment, Faculty of Agriculture, Khon Kaen University, Khon Kaen, Thailand

^c Soil Organic Matter Management Research Group, Khon Kaen University, Khon Kaen, Thailand

^d Department of Microbiology, Faculty of Medicine Khon Kaen University, Khon Kaen, Thailand

^e Department of Land Air and Water Resources, University of California Davis, Davis, California, USA

^f Department of Biochemistry, Faculty of Medicine, Khon Kaen University, Khon Kaen, Thailand

Running head: Biocontrol of *Burkholderia pseudomallei*

#Address correspondence to Rasana W Sermswan, rasana@kku.ac.th

JBG ORCID # orcid.org/0000-0003-1942-7242

RWS ORCID # orcid.org/0000-0002-3691-5064

JLMR ORCID # orcid.org/0000-0002-6446-6462

* Chotima Potisap and Phruaksa Lawongsa contributed equally to this work. The author's order
was determined by those who were hands-on with the work to those who provided ideas and
grant support.

**ABSTRACT**

*Burkholderia pseudomallei*, a saprophytic bacterium responsible for melioidosis, can be
effectively targeted by secondary metabolites produced by *Bacillus amyloliquefaciens* N3-8. In
this study, *B. amyloliquefaciens* N3-8 was applied as a biocontrol agent to sterile soil spiked with
10^7 CFU/g of *B. pseudomallei* at two ratios: 1:10,000 and 1:100,000 CFU/g soil. Both treatments
significantly reduced *B. pseudomallei* by 4-5 logs within four weeks. A subsequent experiment
applied the 1:10,000 ratio to 10 kg of natural soil containing 10^2 - 10^3 CFU/g of *B. pseudomallei*
alongside rice cultivation. Bacterial counts, rice yield, soil physicochemical factors, and microbial
populations were monitored over 110 days. *B. pseudomallei* was undetectable in biocontrol-
treated soil by day 14 but reappeared by day 30, eventually equaling the control soil, suggesting
interference by native microbial communities. No significant differences between the control and
biocontrol treatments were observed in rice yield or soil physicochemical properties.
Metataxonomic analysis revealed 17 bacterial phyla across all samples, with no significant
differences in the overall microbial community structure between treatments at any time point.
On the other hand, significant changes in the microbial beta-diversity over time within the same
soil treatments suggest that temporal dynamics, rather than the biocontrol treatment, drive shifts
in microbial community structure. This study highlights the potential of *B. amyloliquefaciens* N3-
8 as a biocontrol agent against *B. pseudomallei*, and suggests that long-term, repeated
applications are necessary to ensure sustained control with awareness not to disrupt soil microbial
balance.

**IMPORTANCE**

*B. amyloliquefaciens* N3-8 was applied as a biocontrol against *B. pseudomallei*, a pathogenic
bacterium in humans and animals, in soil without significant effect on the physicochemical
properties of soil, the yield of rice model and bacterial structure in soil. However, long-term
treatment is required to obtain sustainable control, and critical management is necessary to avoid
disturbance of the microbial balance in the soil.

**KEYWORDS** biocontrol, melioidosis, rice plantation, soil

**INTRODUCTION**

[revised manuscript text omitted]

**Fig 2.** The number of *B. pseudomallei* in the control and biocontrol soils of the pot model
 with rice plantation. The experiments were done in 4 conditions composed of *B.*
 *pseudomallei* positive soil (C1), *B. pseudomallei* positive soil plus *B. amyloliquefaciens*
 N3-8 at the CFU ratios of 1:10,000 (C2), *B. pseudomallei* positive soil supplemented with

fertilizer (F1), and *B. pseudomallei* positive soil plus *B. amyloliquefaciens* N3-8 at CFU
 ratios of 1:10,000 and supplemented with fertilizer (F2). The soil samples in each pot
 were collected on days 0, 14, 30, 90, and 110 for *B. pseudomallei* culture. The X-axis is
 the days of investigation, and the Y-axis is the number of *B. pseudomallei* in log CFU/g
 soil.

**The effect of *B. amyloliquefaciens* N3-8 on growth and yield of rice.**

The growth (as determined by height and weight) and yield of rice showed no significant
 differences when compared between growing in control soil and biocontrol soil (C1 and C2) and
 between growing in control soil with fertilizer and biocontrol soil with fertilizer (F1 and F2)
 (Table 1).

**Table 1.** The growth and yield of rice in pot model.

Condition	Mean ± SD					
	Height of plant (cm.)	Weight of plant (g)	Panicle number per plant	weight of 100 grains (g)	Good grains /Panicle	Bad grains /Panicle
C1	69.87 ± 0.53	11.23 ± 0.44	3.33 ± 0.28	2.92 ± 0.23	45.67 ± 1.76	21.67 ± 1.45
C2	67.23 ± 0.53	11.15 ± 0.44	3.83 ± 0.28	2.98 ± 0.23	50.33 ± 1.76	18.33 ± 1.45
F1	70.85 ± 0.76	15.45 ± 0.76	5.98 ± 0.50	3.04 ± 0.04	72.10 ± 3.25	16.95 ± 2.51
F2	80.00 ± 0.76	16.76 ± 1.57	6.18 ± 0.89	3.10 ± 0.41	82.68 ± 10.30	13.84 ± 1.74

Wilcoxon Signed Ranks Test compared all parameters between C1 and C2, F1 and F2.

**The effect of *B. amyloliquefaciens* N3-8 on soil physicochemical properties.**

The physicochemical properties of soil were evaluated from pots used for growing rice at
 the following 0, 14, 30, 60, 90, and 110 days. The physicochemical properties of biocontrol soils
 were not significantly different from those measured for control soils at each time point even

though some parameters such as NH_4^+ and NO_3^- fluctuated at some time points as shown in Fig.
 3.

**Fig 3.** Physicochemical properties of soil in rice pot model. The conditions in the pot
 experiments were similar to what was described in Figure 2. The values from the control
 soil (C1) were shown in the black line, the values from the biocontrol soil (C2) were
 shown in the dashed black line, the values in the control soil with fertilizer (F1) were

shown in the gray line, and the values from the biocontrol soil with fertilizer (F2) were
shown in the dash gray line.

**The effect of *B. amyloliquefaciens* N3-8 on the soil bacterial community**

Control soil and biocontrol soil in the rice pot model were collected on days 0, 14, 30, 60,
90, and 110 to analyze the diversity of the bacterial communities by amplification of the V4
region of the 16S rRNA gene and analyzed the microbial metataxonomics (16S rRNA gene-based
metagenomics analysis). All samples, except for F1 and F2 on day 0, yielded adequate reads and
passed the quality parameters, making them suitable for further analysis.

The NMDS plot (Fig. 4) illustrates the differential effects of treatments over time. C1
(represented by circles) and C2 (represented by squares) both exhibit variability within the
NMDS space, indicating changes in microbial community structure over time. Notably, the
microbial communities in all treatments initially cluster together on Days 0, 14, and 30,
demonstrating similar community structures at these time points. After Day 30, a spread is
observed, indicating divergence in community structures over time (Supplementary Fig. 1 and
Fig. 2). However, the PERMANOVA analysis reveals that there are no significant differences in
beta-diversity between the control (C1 and F1) and biocontrol (C2 and F2) indicated as
“Biocontrol” in Table 2 (p-values: C1 vs. C2 = 0.565, F1 vs. F2 = 0.541, and overall = 0.245).
These findings are supported by low R^2 values (0.0171 for C1 vs. C2, 0.0256 for F1 vs. F2, and
0.0384 overall), indicating that the biocontrol treatment explains a small proportion of the
variance in beta-diversity. Contrastingly, when examining differences over time, indicated as
“Days”, within the same soil treatments, significant changes in beta-diversity are evident ($p =$
0.001 for each time comparison), suggesting that temporal dynamics, rather than the biocontrol
treatment per se, are a major driver of changes in microbial community structure (Table 2).

**Fig 4.** Non-metric Multidimensional Scaling (NMDS) plot of the beta-diversity of soil

microbial community at the genus level. This scatter plot graph shows data points

representing different conditions and days, with the axes labeled “NMDS1” and “NMDS2.”

The legend on the right categorizes the data points by treatment (C1, C2, F1, and F2) and

by day (Day 0, Day 14, Day 30, Day 60, Day 90, and Day 110).

**Table 2.** Permutational Multivariate Analysis of Variance in beta-diversity of soil microbial
 community at the genus level.

Data	Days			Biocontrol			Days and Biocontrol		
	R ²	F	p -value	R ²	F	p -value	R ²	F	p -value
Overall	0.3463	5.9910	0.001*	0.0384	1.1197	0.245	0.1408	0.9370	0.676
C1vs C2	0.3965	3.8983	0.001*	0.0171	0.8404	0.565	0.1184	1.1642	0.193
F1vs F2	0.3831	3.4111	0.001*	0.0256	0.9119	0.541	0.0858	0.7640	0.902

*Significant statistical differences at *p*-value <0.05. Distance index: Bray-Curtis.
 R² (Coefficient of Determination): represents the proportion of the total variation in the data that
 is explained by a given factor. The F-value is a ratio used to test the significance of the factors. A
 higher F-value typically indicates a more significant factor.

 The microbial metataxonomics analysis revealed that among the 61 phyla of Bacteria and
 Archaea identified, 17 were consistently present with an average relative abundance above
 0.1% (Figure 5, Supplementary Table 1). All identified phyla belonged to Bacteria, with average
 relative abundances of all observed time points and conditions as follows: Acidobacteriota (13%),
 Actinobacteriota (6%), Armatimonadota (0.5%), Bacteroidota (10.5%), Bdellovibrionota (1.5%),
 Campylobacterota (0.1%), Chloroflexi (5.1%), Cyanobacteria (1.6%), Desulfobacterota (3%),
 Firmicutes (7.7%), Gemmatimonadota (1.9%), Myxococcota (6.7%), Nitrospirota (1.4%),
 Planctomycetota (4.1%), Proteobacteria (30.9%), Spirochaetota (0.5%), and Verrucomicrobiota
 (6.9%) (Figure 5).

**Fig 5.** The relative abundance of microbes in the control and biocontrol soils at the phylum level.

The analysis was done using total DNA extracted from 4 soil conditions: C1 (control soil), C2

(biocontrol soil), F1(control soil with fertilizer), and F2 (biocontrol soil with fertilizer) at 6-time

points of 0, 14, 30, 60, 90 and 110 days.

On Day 0 in the C1 condition, the dominant phyla were Acidobacteriota (24.6%),

followed by Proteobacteria (23.3%), and Chloroflexi (7.6%). Conversely, in the C2 condition on

Day 0, Proteobacteria was the most abundant at 25.4%, with Acidobacteriota at 24.9% and

Myxococcota at 8.3%. Throughout the study, Proteobacteria emerged as the most prevalent

phylum, with its abundance ranging from 20% to 50% across different time points. This phylum
is the largest bacterial phylum and has a variety of roles in the ecosystem (20).

Additionally, the initial samples (Day 0) exhibited higher proportions of Acidobacteriota
and Proteobacteria, whereas samples from Day 14 showed elevated levels of Proteobacteria and
Firmicutes. However, these variations in phylum proportions were consistent across both control
(C1 and F1) and biocontrol (C2 and F2) treatments, demonstrating no significant divergence
between the two experimental setups (Figure 5, Supplementary Table 2 and 3). This consistency
underscores the uniformity in microbial community structure regardless of the treatment applied.

**DISCUSSION**

Soil and water are known to be the important reservoirs of *B. pseudomallei* that lead to
infection in humans and animals when come into contact with contaminated ones (21). The
bacterium can survive through the dry season and nutrient depletion conditions perhaps by
biofilm protection and then expand during the rainy season (9). A large number of patients
infected with *B. pseudomallei* are rice agriculturists (22). Controlling the pathogens in soil
should be beneficial to those who are at risk.

In a laboratory scale experiment, a biocontrol treatment using *B. amyloliquefaciens* N3-8
of either 1:10,000 or 1:100,000 CFU ratio, with or without another addition of *B.*
*amyloliquefaciens* N3-8, could significantly reduce the pathogen in sterile soil spiked with a
known amount of *B. pseudomallei*. However, when the 1:10,000 ratio was applied in the rice pot
model, it could effectively decrease the pathogen until undetected within 14 days, but the
pathogen became detectable in 30 days. Thereafter, the number of this pathogen was increased
both in the control and the biocontrol soil. The amount of *B. pseudomallei* may fluctuate during

the study but in a similar pattern between the control (C1) and fertilizer control (F1) conditions.
Better control of *B. pseudomallei* observed on a small laboratory scale may be due to the presence
of secondary metabolites produced by *B. amyloliquefaciens* N3-8, causing the killing of the
pathogen. Moreover, a sterile soil condition could abolish the influences of other organisms in the
soil. Effective control of the pathogen in natural soil conditions could mainly come from a
combination of factors that may not or inferior support the growth or activity of *B.*
*amyloliquefaciens* N3-8. This idea was supported by the fact that we were previously able to
isolate the biocontrol agent and other *B. amyloliquefaciens* isolates with activity to kill this
pathogen from soils where *B. pseudomallei* was not detected (15, 16). Consistent application of
the biocontrol or the identification of some soil physicochemical properties in natural soil that
support *B. amyloliquefaciens* may help to sustain the biocontrol process. A report from Australia
indicated the association of *B. pseudomallei* in the rhizosphere and the roots of specific grasses
(23). As rice is also a monocotyledon, this association could make the killing less effective and
may require more than one-time application of the biocontrol agent. A more in-depth observation
of the association of the pathogen with the rice rhizosphere is needed and it may help in designing
more effective biocontrol strategies in the future.

Some soil chemical parameters, such as total N and NO_3^- fluctuated at some time points in
both control and biocontrol treatments. Nitrogen (N) is one of the most important inputs used to
increase plant growth and productivity, and nitrate (NO_3^-) is a form of nitrogen that plants can use
to produce their essential molecules, such as chlorophyll and proteins. Several forms of N can be
detected in flooded rice soils, and it can be lost through ammonia (NH_3) volatilization, and

[revised manuscript text omitted]

for 15s), annealing (50°C for 30s), and elongation (72°C for 15s), followed by a final extension
(72°C for 10min). Paired-end sequencing, with 250 bp reads, was performed using the Miseq
platform (Illumina, Inc.) at the University of California-Davis Genomics Facility, Davis, CA,
according to the manufacturer's instructions.

**Bioinformatics and statistical analyses**

All bioinformatics and statistical analyses were performed on RStudio 4.3.1(43). Raw
sequences were processed by inferring the amplicon sequence variants (ASVs) using the Dada2
1.28.0 package(44). We obtained 4,378,031 sequencing reads. Forward and reverse reads with a
phred score < 30 were truncated at position 245 and 180 base pair, respectively, and processed for
filtering, trimming, error correction, dereplication, merge, and chimera removal. After quality
control, a total of 2,806,526 microbial sequences with an average length of 225 bp were obtained.
Taxonomy was assigned using the SILVA reference database (release 138.1, 07.03.2021). The
ASV counts classified in the same taxonomic groups were summarized and normalized to within-
sample relative abundance following the total sum scaling method (TTS) (45). Microbial taxa
were grouped at the phylum and genus levels for further analysis.

Statistical analyses and graphical visualization were performed using vegan 2.6–4
(Oksanen et al., 2022), ARTool 0.11.1 (Kay et al., 2021), dplyr 1.1.3 (Wickham et al., 2023),
tidyr 1.3.0 (Wickham et al., 2023), scales 1.2.1(Wickham et al., 2023), pheatmap 1.0.12 (Kolde,
2019), psych 2.3.9 (Revelle, 2023), lsmean 2.30-0 (Lenth, 2018) and ggplot2 3.4.3 (Wickham et
al., 2023) packages. Nonmetric multidimensional scaling (NMDS) and permutational multivariate
analysis of variance (PERMANOVA) followed by FDR adjustment were used to estimate the
similarities among different treatments (Bray-Curtis distance) at the genus level. Two-way
ANOVA of aligned rank transformed data was used to investigate the effect of each treatment

and sampling day on the relative abundances of the most abundant Phyla (>0.1%), then followed
by pairwise comparisons using FDR adjustment.

**ACKNOWLEDGMENTS**

Dr. Chotima Potisap was supported by Post-Doctoral Training PD2564-08, Khon Kean
University. This project is funded by the National Research Council of Thailand (NRCT).

**REFERENCES**

[revised manuscript text omitted]

- 36. Samosornsuk N, Lulitanond A, Saenla N, Anuntagool N, Wongratanacheewin S, Sirisinha
S. 1999. Short report: evaluation of a monoclonal antibody-based latex agglutination test
for rapid diagnosis of septicemic melioidosis. *Am J Trop Med Hyg* 61:735-7.
- 37. Dewis J, Freitas F. 1970. Physical and chemical methods of soil and water analysis. FAO
Soils Bulletin:275 pp.
- 38. Kamara A, Rhodes ER, Sawyerr PA. 2007. Dry Combustion Carbon, Walkley–Black
Carbon, and Loss on Ignition for Aggregate Size Fractions on a Toposequence.
*Communications in Soil Science and Plant Analysis* 38:2005-2012.
- 39. Bremner JM. 1965. Total Nitrogen, p 1149-1178, *Methods of Soil Analysis*
doi:<https://doi.org/10.2134/agronmonogr9.2.c32>.
- 40. Keeney DR, Nelson DW. 1982. Nitrogen—Inorganic Forms, p 643-698, *Methods of Soil*
*Analysis* doi:<https://doi.org/10.2134/agronmonogr9.2.2ed.c33>.
- 41. Parada AE, Needham DM, Fuhrman JA. 2016. Every base matters: assessing small
subunit rRNA primers for marine microbiomes with mock communities, time series and
global field samples. *Environ Microbiol* 18:1403-14.
- 42. Walters W, Hyde ER, Berg-Lyons D, Ackermann G, Humphrey G, Parada A, Gilbert JA,
Jansson JK, Caporaso JG, Fuhrman JA, Apprill A, Knight R. 2016. Improved Bacterial
16S rRNA Gene (V4 and V4-5) and Fungal Internal Transcribed Spacer Marker Gene
Primers for Microbial Community Surveys. *mSystems* 1.
- 43. Anonymous. 2023. R CORE Teame . A language and environment for statistical
computing. , R Foundation for Statistical Computing<https://www.R-project.org>, Vienna,
Austria.
- 44. Callahan BJ, McMurdie PJ, Rosen MJ, Han AW, Johnson AJ, Holmes SP. 2016. DADA2:
High-resolution sample inference from Illumina amplicon data. *Nat Methods* 13:581-3.

45. Lin H, Peddada SD. 2020. Analysis of microbial compositions: a review of normalization
and differential abundance analysis. NPJ Biofilms Microbiomes 6:60.

Response to reviewers

Spectrum02963-24 (The soil microorganism *Bacillus amyloliquefaciens* N3-8 is a biocontrol agent for the pathogen *Burkholderia pseudomallei* in rice plantation)

Reviewer #1 (Public repository details (Required)):

The authors did not indicate in their manuscript if they deposited their metagenomic data in a repository.

Answer: Thank you very much for your comment. The 16S rRNA sequencing dataset generated and analyzed during the current study is available in NCBI's Sequence Read Archive (SRA) under the accession number PRJNA1215386. This was mentioned in the main text.

Reviewer #1 (Comments for the Author):

The document is about the use of the soil microorganism *Bacillus amyloliquefaciens* N3-8 as a biocontrol agent for the pathogen *Burkholderia pseudomallei* in rice plantations to prevent melioidosis. *Burkholderia pseudomallei*, the causative agent of melioidosis, poses significant health risks, particularly in Southeast Asia and Northern Australia. The authors identified *Bacillus amyloliquefaciens* N3-8 as a potential biocontrol agent against *B. pseudomallei* and evaluated the effectiveness of *B. amyloliquefaciens* N3-8 in reducing *B. pseudomallei* proliferation in both controlled laboratory settings and more naturalistic pot experiments with rice plants. However, the paper needs thorough language editing.

Answer: The language of the manuscript was thoroughly edited as suggested.

Major comments

1. The authors should showcase the limitations of the study in the discussion and abstract as this will help justify the methods they employed in the study.

Answer: Thank you very much for pointing this out. The study's limitations that performed the experiments in the pot scale were mentioned in the abstract and discussion as suggested.

2. From the authors conclusions the biocontrol treatment did not significantly affect the rice yield; showed no significant differences when compared between the control soil and biocontrol soil (C1 and C2) and between the control soil with fertilizer and biocontrol soil with fertilizer (F1 and F2). Invariably the biocontrol agent had no plant growth-promoting properties. It would have provided some clarity if authors characterized the secondary metabolite secreted by the bacillus strain to ascertain it had no plant growth-promoting properties. As it is rare for *Bacillus* strain not to possess both biocontrol and PGP properties in tandem, the authors should provide past work/research to justify the conclusions that biocontrol agent could only antagonize *B. pseudomallei* and not promote plant growth simultaneously.

Answer: I agree that *Bacillus* strains often show PGP properties in various plants, including rice. However, we did not detect a difference in the growth or productivity of rice that was

supplemented with *Bacillus amyloliquefaciens* in this study. This may be due to the short duration of treatment or condition in the pot model. We are characterizing the secondary metabolites of the N3-8 strain and will focus more on its PGP properties. We have discussed these points more in the discussion.

3. Figures: In general, the figure legends need clearer information and should be formatted per the journal requirements as well as up to the standard required for publication.

Answer: Thank you for your suggestions. The figure legends were fixed and formatted for revision.

Other minor comments can be found in the attached PDF file

All comments in the PDF file, including the title of the manuscript, were edited or rephrased according to the reviewers' comments.

Thanks for all your suggestions.

Reviewer #2 (Comments for the Author):

Review on " The soil microorganism *Bacillus amyloliquefaciens* N3-8 is a biocontrol agent for the pathogen *Burkholderia pseudomallei* in rice plantation" by Potisap et al. for Microbiology Spectrum Journal.

In this article the authors have evaluated the impact of co-inoculating a *Bacillus amyloliquefaciens* strain (shown previously to antagonize *Burkholderia pseudomallei* in laboratory) in a soil containing a defined concentration of *B. pseudomallei* cells. They monitored the number of *B. pseudomallei* at different times post inoculation, and monitored the impact during a rice cycle in a pot experiments, on *B. pseudomallei* counts, soil physico chemical structure, rice growth parameters, and bacterial community structure (using 16S V4 amplicon barcoding). The *Bacillus* inoculation reduced the CFU of *B. pseudomallei* in soil but its impact did not eradicate the pathogen which growth kinetics increased at later times. The *Bacillus* inoculation had no statistically supported impact on rice growth parameters, and the bacterial communities changed over time but authors did not look into details at genus level to see enrichment or depletion between conditions. Globally the study is interesting and the topic to use bacterial inoculant to biocontrol human pathogens in soils is interesting. I have made several major and minor comments to improve the manuscript.

Major comments:

1. The title of the manuscript is dedicated to biocontrol against *B. pseudomallei*, but half of the work is also dedicated to monitor the impact of *Bacillus* inoculation on rice productivity. The title and manuscript should better integrate the two dimensions of the potential of *Bacillus* inoculation: biocontrol and plant growth promoting properties on rice

Answer: Thank you for your comment. The title was edited as suggested.

2. The control used to compare bacillus inoculation in soil against *B. pseudomallei* is soil without inoculation. A good control would have been a bacterial strain with no known PGPR or biocontrol effect, as an *E. coli* for example. Because the fact to inoculate a number of bacterial cells in exponential phase in high numbers can have an impact whatever their biocontrol capacities. It could have revealed the specificity and particularities of *Bacillus* compared to others.

Answer: Thank you very much for your interesting suggestion. Even though we cannot incorporate the idea this time, we will try this concept in our next experiment.

3. The 16S amplicon barcoding approach could be used also to monitor *B. pseudomallei* counts, even if the V4 region of 16S is not highly specific, I guess the authors have not looked at *B. pseudomallei* counts in their ASV count table. Did they identify the *B. pseudomallei* or *Burkholderia* genus? These data could be interesting to look at. On the other hand, given the authors have the DNA for metagenomics, they could use PCR or qPCR to identify and quantify *B. pseudomallei* in the rice pot experiment. There are qPCR primers described in the literature to quantify *B. pseudomallei* cells.

Answer: Thank you for your insightful comment regarding the identification and quantification of *B. pseudomallei*. To address your comment, we have included Fig. 6 (lines 200-206), which illustrates the variation in the relative abundance of the *Burkholderia* genus across treatments throughout the experiment. Our analysis revealed that *Burkholderia* was present at low abundance across all treatments and time points, with some samples showing it as undetectable. Considering the conditions of our experiment, the *Bacillus* present in C1 and F1 were present in soil naturally, while those in C2 and F2 were from natural plus *B. amyloliquefaciens* N3-8. Moreover, the number of *B. pseudomallei* in C2 and F2 should be composed of both live and dead as affected by the biocontrol agent.

When biocontrol of *B. pseudomallei* was focused on in our experiments, we would like to count the number of live bacteria. The specific qPCR will amplify remnants of their DNA in the soil even if they were killed. We acknowledge its potential for providing more accurate species-level data. Thank you for this valuable recommendation.

4. Authors should evaluate genus enrichments or depletion across conditions in their 16S amplicon barcoding data. If no main differences are observed (apart from time) globally it does not mean that specific genera (or even species) cannot be detected using specific tests (Wilcoxon tests for examples). It could be interesting at least to look at the *Burkholderia* and *Bacillus* genera in particular, given these have been inoculated (or not) between conditions.

Answer: As suggested, we reanalyzed the data, filtering ASV counts at the genus level for *Burkholderia* and *Bacillus* (Fig. 6). The relative abundance of *Bacillus* was significantly higher across all treatments compared to *Burkholderia*, which was undetectable on certain days. To further investigate potential genus-level differences, we performed a Wilcoxon rank-sum test to compare the relative abundance of these genera across treatments, but no statistically significant enrichment or depletion was detected. Additionally, we conducted a linear regression analysis to

assess potential correlations between *Bacillus* and *Burkholderia* across treatments, which also showed no significant association. Given that 16S sequencing data is compositional, we emphasize that inoculation with *Bacillus amyloliquefaciens* did not significantly alter the overall microbial community structure. Notably, by Day 14 post-inoculation, *B. pseudomallei* was no longer culturable on Ashdown's selective medium, indicating a potential shift in its viability rather than its relative abundance in sequencing data. For reference, we have incorporated these new results in lines 200-206 of the manuscript.

5. The part on the impact of inoculation on soil physicochemical parameters should be shortened and main results better underlined, and shortened in the discussion while focusing on main impact (which appears to be on nitrogen, but why authors did focus on nitrogen ? and not other nutrients?).

Answer: Thank you for your suggestion. We adjusted the content as suggested. We mentioned only the nitrogen because it was changed in a broader range than others; however, it was within the acceptable ranges for rice cultivation as other parameters.

Minor comments

In all the manuscript, authors give the strain name of *Bacillus*, while they never give it for *B. pseudomallei* except in the material and methods. Authors should indicate in the text everywhere they mention *B. pseudomallei* that it is strain p37.

Answer: The name of the strain p37 was added where appropriate. Thank you.

Line 48 to 52: rephrase to "B. amyloliquefaciens N3-8 has been used in soil as a biocontrol agent against *B. pseudomallei*, a bacterium pathogenic to humans and animals, where it has shown significant effects on soil physicochemical properties, rice yield and bacterial community structure. However, long-term treatments are needed to achieve sustainable control and critical management is required to avoid disturbing the microbial balance in the soil"

Answer: Lines 48-52: The sentences were rephrased as kindly suggested.

Line 107: "autoclaved sterile": change to : sterilized by autoclaving . Authors should indicate conditions of autoclaving (one or 2 times, which temperature and pressure), as soil autoclaving conditions can be critical in achieving good sterilization.

Answer: The words were changed in the methods and results sections, and details were added.

Line 158: "The effect of *B. amyloliquefaciens* N3-8 on growth and yield of rice". I don't understand if there is an effect of fertilization or not on rice yield? The table 1 does not indicate any significant impact of either fertilization or bioinoculation on rice yield. If this is the case it should be clearly written in the text that *Bacillus* inoculation did not increase any rice growth parameters in the tested conditions.

Answer: Thank you very much for your comments. Fertilizers do affect rice growth and yield. However, we did not show the statistical analysis of them on the growth and yield of rice (C1 VS F1 and C2 VS F2). We focus on whether the biocontrol agent has any effects on the growth and yield of rice by comparing C1 and C2 or F1 and F2. There were no significant differences in any parameters. To make the results clearer to readers, we explained our idea more in the text (Lines 135-140).

Line 188: The authors should be careful when describing their NMDS plot, as they did mention differences in the text but apart from a time impact on beta diversity, there is not statistical difference between treatments when they tested it in Table 2.

Answer: Thank you for pointing this out. Lines 158-171: We have revised the paragraph in question to clarify that while the NMDS plot visually suggests differences, the statistical analysis via PERMANOVA does not support significant differences in beta-diversity between the control and biocontrol treatments. The revised text explicitly emphasizes the lack of statistical significance (p-values and R² values from Table 2) and highlights that temporal dynamics are the main driver of changes in microbial community structure, as supported by the statistical analysis. We believe this revision addresses your concern and aligns the narrative more closely with the statistical findings.

Line 235: Figure 5: in this figure authors should include at least a binning plot at genus level to monitor Burkholderias. Showing diversity binning plots at order or class level is not informative at all. To detect differences between treatments, authors could have performed enrichment tests to see at genus level if there are enriched genera in the treatments, using Kruskal-Wallis or Wilcoxon tests. This is common tests in amplicon barcoding analyses.

Thank you for your suggestion. As recommended, we have included a binning plot at the genus level to specifically monitor *Burkholderia* in Fig. 6, providing a more detailed view of its relative abundance across treatments. To further assess potential differences between treatments, we performed Wilcoxon tests at the genus level. However, we did not detect any statistically significant enrichment or depletion of *Burkholderia*, likely due to its very low abundance, with some sampling points showing it as undetectable. Nonetheless, our results are particularly relevant as they demonstrate that the inoculation of *B. amyloliquefaciens* effectively controlled *B. pseudomallei* by Day 14 when it became undetectable on Ashdown's selective medium. Importantly, *B. amyloliquefaciens* did not significantly alter the overall microbial community structure. For reference, we have updated the manuscript accordingly (lines 200-206).

Line 268-269 ; "Better control of *B. pseudomallei* observed on a small laboratory scale may be due to the presence of secondary metabolites produced by *B. amyloliquefaciens* N3-8, causing the killing of the pathogen": this is purely speculative, it could be the result of many different other things, if authors want to test this hypothesis they should make experiments to reveal secondary metabolites production in their pot condition, or at least give insights in the discussion on how to test it to prove it.

Answer: Thank you for your valuable suggestion. The characterization of *B. amyloliquefaciens* N3-8 secondary metabolites has just started. We rewrote and gave some discussion according to this point in the text (Lines 222-229).

Line 357-361: direct detection of *B. pseudomallei*. Why no molecular methods were used to confirm the belonging of the strains to *B. pseudomallei* species? What is the robustness and specificity of the latex agglutination method? Specific primers do exist to identify *B. pseudomallei*.

Answer: A direct culture of *B. pseudomallei* on Ashdown's selective medium was used to observe the presence of the live pathogen after biocontrol treatment. Latex agglutination can be used to confirm each colony from the plates. The method is very quick and highly specific (Am J Trop Med Hyg. 2014 Jun 4;90(6):1043–1046). The PCR test to confirm each colony of *B. pseudomallei* will take more time in our setting.

Line 339-344 (conclusion): The results do not show a strong decrease of *B. pseudomallei* in the long term, authors should be more cautious on their conclusion to no oversell the impact of inoculation. There are no statistically significant impact of the inoculation on any rice growth parameters, and biocontrol of *B. pseudomallei* did not last.

Answer: Thank you for your comments. The conclusion was rewritten to be more precise to our findings.

3

4 Chotima Potisap^{a*}, Phrueksa Lawongsa^{b,c*}, Jittima Duangsri^{a,d}, Júlia B. Gontijo^e, Surasakdi
5 Wongratanacheewin^{a,d}, Jorge L Mazza Rodrigues^e, Rasana W Sermswan^{a,f,#}

6

7 ^a Melioidosis Research Center, Khon Kaen University, Khon Kaen, Thailand

8 ^b Department of Soil Science and Environment, Faculty of Agriculture, Khon Kaen University,
9 Khon Kaen, Thailand

10 ^c Soil Organic Matter Management Research Group, Khon Kaen University, Khon Kaen,
11 Thailand

12 ^d Department of Microbiology, Faculty of Medicine Khon Kaen University, Khon Kaen, Thailand

13 ^e Department of Land Air and Water Resources, University of California Davis, Davis,
14 California, USA

15 ^f Department of Biochemistry, Faculty of Medicine, Khon Kaen University, Khon Kaen, Thailand

Running head: Biocontrol of *Burkholderia pseudomallei*

#Address correspondence to Rasana W Sermswan, rasana@kku.ac.th

JBG ORCID # orcid.org/0000-0003-1942-7242

RWS ORCID # orcid.org/0000-0002-3691-5064

JLMR ORCID # orcid.org/0000-0002-6446-6462

* Chotima Potisap and Phruaksa Lawongsa contributed equally to this work. The author's order
was determined by those who were hands-on with the work to those who provided ideas and
grant support.

**ABSTRACT**

*Burkholderia pseudomallei*, a saprophytic bacterium responsible for melioidosis, can be
effectively targeted by secondary metabolites produced by *Bacillus amyloliquefaciens* N3-8. In
this study, *B. amyloliquefaciens* N3-8 was applied as a biocontrol agent to sterile soil spiked with
10^7 CFU/g of *B. pseudomallei* at two ratios: 1:10,000 and 1:100,000 CFU/g soil. Both treatments
significantly reduced *B. pseudomallei* by 4-5 logs within four weeks. A subsequent experiment
applied the 1:10,000 ratio to 10 kg of natural soil containing 10^2 - 10^3 CFU/g of *B. pseudomallei*
alongside rice cultivation. Bacterial counts, rice yield, soil physicochemical factors, and microbial
populations were monitored over 110 days. *B. pseudomallei* was undetectable in biocontrol-
treated soil by day 14 but reappeared by day 30, eventually equaling the control soil, suggesting
interference by native microbial communities. No significant differences between the control and
biocontrol treatments were observed in rice yield or soil physicochemical properties.
Metataxonomic analysis revealed 17 bacterial phyla across all samples, with no significant
differences in the overall microbial community structure between treatments at any time point.
On the other hand, significant changes in the microbial beta-diversity over time within the same
soil treatments suggest that temporal dynamics, rather than the biocontrol treatment, drive shifts
in microbial community structure. This study highlights the potential of *B. amyloliquefaciens* N3-
8 as a biocontrol agent against *B. pseudomallei*, and suggests that long-term, repeated
applications are necessary to ensure sustained control with awareness not to disrupt soil microbial
balance.

**IMPORTANCE**

*B. amyloliquefaciens* N3-8 was applied as a biocontrol against *B. pseudomallei*, a pathogenic
bacterium in humans and animals, in soil without significant effect on the physicochemical
properties of soil, the yield of rice model and bacterial structure in soil. However, long-term
treatment is required to obtain sustainable control, and critical management is necessary to avoid
disturbance of the microbial balance in the soil.

**KEYWORDS** biocontrol, melioidosis, rice plantation, soil

**INTRODUCTION**

[revised manuscript text omitted]

**Fig 2.** The number of *B. pseudomallei* in the control and biocontrol soils of the pot model
 with rice plantation. The experiments were done in 4 conditions composed of *B.*
 *pseudomallei* positive soil (C1), *B. pseudomallei* positive soil plus *B. amyloliquefaciens*
 N3-8 at the CFU ratios of 1:10,000 (C2), *B. pseudomallei* positive soil supplemented with

fertilizer (F1), and *B. pseudomallei* positive soil plus *B. amyloliquefaciens* N3-8 at CFU
 ratios of 1:10,000 and supplemented with fertilizer (F2). The soil samples in each pot
 were collected on days 0, 14, 30, 90, and 110 for *B. pseudomallei* culture. The X-axis is
 the days of investigation, and the Y-axis is the number of *B. pseudomallei* in log CFU/g
 soil.

**The effect of *B. amyloliquefaciens* N3-8 on growth and yield of rice.**

The growth (as determined by height and weight) and yield of rice showed no significant
 differences when compared between growing in control soil and biocontrol soil (C1 and C2) and
 between growing in control soil with fertilizer and biocontrol soil with fertilizer (F1 and F2)
 (Table 1).

**Table 1.** The growth and yield of rice in pot model.

Condition	Mean ± SD					
	Height of plant (cm.)	Weight of plant (g)	Panicle number per plant	weight of 100 grains (g)	Good grains /Panicle	Bad grains /Panicle
C1	69.87 ± 0.53	11.23 ± 0.44	3.33 ± 0.28	2.92 ± 0.23	45.67 ± 1.76	21.67 ± 1.45
C2	67.23 ± 0.53	11.15 ± 0.44	3.83 ± 0.28	2.98 ± 0.23	50.33 ± 1.76	18.33 ± 1.45
F1	70.85 ± 0.76	15.45 ± 0.76	5.98 ± 0.50	3.04 ± 0.04	72.10 ± 3.25	16.95 ± 2.51
F2	80.00 ± 0.76	16.76 ± 1.57	6.18 ± 0.89	3.10 ± 0.41	82.68 ± 10.30	13.84 ± 1.74

Wilcoxon Signed Ranks Test compared all parameters between C1 and C2, F1 and F2.

**The effect of *B. amyloliquefaciens* N3-8 on soil physicochemical properties.**

The physicochemical properties of soil were evaluated from pots used for growing rice at
 the following 0, 14, 30, 60, 90, and 110 days. The physicochemical properties of biocontrol soils
 were not significantly different from those measured for control soils at each time point even

though some parameters such as NH_4^+ and NO_3^- fluctuated at some time points as shown in Fig.
 3.

**Fig 3.** Physicochemical properties of soil in rice pot model. The conditions in the pot
 experiments were similar to what was described in Figure 2. The values from the control
 soil (C1) were shown in the black line, the values from the biocontrol soil (C2) were
 shown in the dashed black line, the values in the control soil with fertilizer (F1) were

shown in the gray line, and the values from the biocontrol soil with fertilizer (F2) were
shown in the dash gray line.

**The effect of *B. amyloliquefaciens* N3-8 on the soil bacterial community**

Control soil and biocontrol soil in the rice pot model were collected on days 0, 14, 30, 60,
90, and 110 to analyze the diversity of the bacterial communities by amplification of the V4
region of the 16S rRNA gene and analyzed the microbial metataxonomics (16S rRNA gene-based
metagenomics analysis). All samples, except for F1 and F2 on day 0, yielded adequate reads and
passed the quality parameters, making them suitable for further analysis.

The NMDS plot (Fig. 4) illustrates the differential effects of treatments over time. C1
(represented by circles) and C2 (represented by squares) both exhibit variability within the
NMDS space, indicating changes in microbial community structure over time. Notably, the
microbial communities in all treatments initially cluster together on Days 0, 14, and 30,
demonstrating similar community structures at these time points. After Day 30, a spread is
observed, indicating divergence in community structures over time (Supplementary Fig. 1 and
Fig. 2). However, the PERMANOVA analysis reveals that there are no significant differences in
beta-diversity between the control (C1 and F1) and biocontrol (C2 and F2) indicated as
“Biocontrol” in Table 2 (p-values: C1 vs. C2 = 0.565, F1 vs. F2 = 0.541, and overall = 0.245).
These findings are supported by low R^2 values (0.0171 for C1 vs. C2, 0.0256 for F1 vs. F2, and
0.0384 overall), indicating that the biocontrol treatment explains a small proportion of the
variance in beta-diversity. Contrastingly, when examining differences over time, indicated as
“Days”, within the same soil treatments, significant changes in beta-diversity are evident ($p =$
0.001 for each time comparison), suggesting that temporal dynamics, rather than the biocontrol
treatment per se, are a major driver of changes in microbial community structure (Table 2).

**Fig 4.** Non-metric Multidimensional Scaling (NMDS) plot of the beta-diversity of soil
 microbial community at the genus level. This scatter plot graph shows data points
 representing different conditions and days, with the axes labeled “NMDS1” and “NMDS2.”
 The legend on the right categorizes the data points by treatment (C1, C2, F1, and F2) and
 by day (Day 0, Day 14, Day 30, Day 60, Day 90, and Day 110).

**Table 2.** Permutational Multivariate Analysis of Variance in beta-diversity of soil microbial
 community at the genus level.

Data	Days			Biocontrol			Days and Biocontrol		
	R ²	F	p -value	R ²	F	p -value	R ²	F	p -value
Overall	0.3463	5.9910	0.001*	0.0384	1.1197	0.245	0.1408	0.9370	0.676
C1vs C2	0.3965	3.8983	0.001*	0.0171	0.8404	0.565	0.1184	1.1642	0.193
F1vs F2	0.3831	3.4111	0.001*	0.0256	0.9119	0.541	0.0858	0.7640	0.902

*Significant statistical differences at *p*-value <0.05. Distance index: Bray-Curtis.
 R² (Coefficient of Determination): represents the proportion of the total variation in the data that
 is explained by a given factor. The F-value is a ratio used to test the significance of the factors. A
 higher F-value typically indicates a more significant factor.

 The microbial metataxonomics analysis revealed that among the 61 phyla of Bacteria and
 Archaea identified, 17 were consistently present with an average relative abundance above
 0.1% (Figure 5, Supplementary Table 1). All identified phyla belonged to Bacteria, with average
 relative abundances of all observed time points and conditions as follows: Acidobacteriota (13%),
 Actinobacteriota (6%), Armatimonadota (0.5%), Bacteroidota (10.5%), Bdellovibrionota (1.5%),
 Campylobacterota (0.1%), Chloroflexi (5.1%), Cyanobacteria (1.6%), Desulfobacterota (3%),
 Firmicutes (7.7%), Gemmatimonadota (1.9%), Myxococcota (6.7%), Nitrospirota (1.4%),
 Planctomycetota (4.1%), Proteobacteria (30.9%), Spirochaetota (0.5%), and Verrucomicrobiota
 (6.9%) (Figure 5).

**Fig 5.** The relative abundance of microbes in the control and biocontrol soils at the phylum level.

The analysis was done using total DNA extracted from 4 soil conditions: C1 (control soil), C2

(biocontrol soil), F1(control soil with fertilizer), and F2 (biocontrol soil with fertilizer) at 6-time

points of 0, 14, 30, 60, 90 and 110 days.

On Day 0 in the C1 condition, the dominant phyla were Acidobacteriota (24.6%),

followed by Proteobacteria (23.3%), and Chloroflexi (7.6%). Conversely, in the C2 condition on

Day 0, Proteobacteria was the most abundant at 25.4%, with Acidobacteriota at 24.9% and

Myxococcota at 8.3%. Throughout the study, Proteobacteria emerged as the most prevalent

phylum, with its abundance ranging from 20% to 50% across different time points. This phylum
is the largest bacterial phylum and has a variety of roles in the ecosystem (20).

Additionally, the initial samples (Day 0) exhibited higher proportions of Acidobacteriota
and Proteobacteria, whereas samples from Day 14 showed elevated levels of Proteobacteria and
Firmicutes. However, these variations in phylum proportions were consistent across both control
(C1 and F1) and biocontrol (C2 and F2) treatments, demonstrating no significant divergence
between the two experimental setups (Figure 5, Supplementary Table 2 and 3). This consistency
underscores the uniformity in microbial community structure regardless of the treatment applied.

**DISCUSSION**

Soil and water are known to be the important reservoirs of *B. pseudomallei* that lead to
infection in humans and animals when come into contact with contaminated ones (21). The
bacterium can survive through the dry season and nutrient depletion conditions perhaps by
biofilm protection and then expand during the rainy season (9). A large number of patients
infected with *B. pseudomallei* are rice agriculturists (22). Controlling the pathogens in soil
should be beneficial to those who are at risk.

In a laboratory scale experiment, a biocontrol treatment using *B. amyloliquefaciens* N3-8
of either 1:10,000 or 1:100,000 CFU ratio, with or without another addition of *B.*
*amyloliquefaciens* N3-8, could significantly reduce the pathogen in sterile soil spiked with a
known amount of *B. pseudomallei*. However, when the 1:10,000 ratio was applied in the rice pot
model, it could effectively decrease the pathogen until undetected within 14 days, but the
pathogen became detectable in 30 days. Thereafter, the number of this pathogen was increased
both in the control and the biocontrol soil. The amount of *B. pseudomallei* may fluctuate during

the study but in a similar pattern between the control (C1) and fertilizer control (F1) conditions.
Better control of *B. pseudomallei* observed on a small laboratory scale may be due to the presence
of secondary metabolites produced by *B. amyloliquefaciens* N3-8, causing the killing of the
pathogen. Moreover, a sterile soil condition could abolish the influences of other organisms in the
soil. Effective control of the pathogen in natural soil conditions could mainly come from a
combination of factors that may not or inferior support the growth or activity of *B.*
*amyloliquefaciens* N3-8. This idea was supported by the fact that we were previously able to
isolate the biocontrol agent and other *B. amyloliquefaciens* isolates with activity to kill this
pathogen from soils where *B. pseudomallei* was not detected (15, 16). Consistent application of
the biocontrol or the identification of some soil physicochemical properties in natural soil that
support *B. amyloliquefaciens* may help to sustain the biocontrol process. A report from Australia
indicated the association of *B. pseudomallei* in the rhizosphere and the roots of specific grasses
(23). As rice is also a monocotyledon, this association could make the killing less effective and
may require more than one-time application of the biocontrol agent. A more in-depth observation
of the association of the pathogen with the rice rhizosphere is needed and it may help in designing
more effective biocontrol strategies in the future.

Some soil chemical parameters, such as total N and NO_3^- fluctuated at some time points in
both control and biocontrol treatments. Nitrogen (N) is one of the most important inputs used to
increase plant growth and productivity, and nitrate (NO_3^-) is a form of nitrogen that plants can use
to produce their essential molecules, such as chlorophyll and proteins. Several forms of N can be
detected in flooded rice soils, and it can be lost through ammonia (NH_3) volatilization, and

[revised manuscript text omitted]

for 15s), annealing (50°C for 30s), and elongation (72°C for 15s), followed by a final extension
(72°C for 10min). Paired-end sequencing, with 250 bp reads, was performed using the Miseq
platform (Illumina, Inc.) at the University of California-Davis Genomics Facility, Davis, CA,
according to the manufacturer's instructions.

**Bioinformatics and statistical analyses**

All bioinformatics and statistical analyses were performed on RStudio 4.3.1(43). Raw
sequences were processed by inferring the amplicon sequence variants (ASVs) using the Dada2
1.28.0 package(44). We obtained 4,378,031 sequencing reads. Forward and reverse reads with a
phred score < 30 were truncated at position 245 and 180 base pair, respectively, and processed for
filtering, trimming, error correction, dereplication, merge, and chimera removal. After quality
control, a total of 2,806,526 microbial sequences with an average length of 225 bp were obtained.
Taxonomy was assigned using the SILVA reference database (release 138.1, 07.03.2021). The
ASV counts classified in the same taxonomic groups were summarized and normalized to within-
sample relative abundance following the total sum scaling method (TTS) (45). Microbial taxa
were grouped at the phylum and genus levels for further analysis.

Statistical analyses and graphical visualization were performed using vegan 2.6-4
(Oksanen et al., 2022), ARTool 0.11.1 (Kay et al., 2021), dplyr 1.1.3 (Wickham et al., 2023),
tidyr 1.3.0 (Wickham et al., 2023), scales 1.2.1(Wickham et al., 2023), pheatmap 1.0.12 (Kolde,
2019), psych 2.3.9 (Revelle, 2023), lsmean 2.30-0 (Lenth, 2018) and ggplot2 3.4.3 (Wickham et
al., 2023) packages. Nonmetric multidimensional scaling (NMDS) and permutational multivariate
analysis of variance (PERMANOVA) followed by FDR adjustment were used to estimate the
similarities among different treatments (Bray-Curtis distance) at the genus level. Two-way
ANOVA of aligned rank transformed data was used to investigate the effect of each treatment

and sampling day on the relative abundances of the most abundant Phyla (>0.1%), then followed
by pairwise comparisons using FDR adjustment.

**ACKNOWLEDGMENTS**

Dr. Chotima Potisap was supported by Post-Doctoral Training PD2564-08, Khon Kean
University. This project is funded by the National Research Council of Thailand (NRCT).

**REFERENCES**

[revised manuscript text omitted]

- 36. Samosornsuk N, Lulitanond A, Saenla N, Anuntagool N, Wongratanacheewin S, Sirisinha
S. 1999. Short report: evaluation of a monoclonal antibody-based latex agglutination test
for rapid diagnosis of septicemic melioidosis. *Am J Trop Med Hyg* 61:735-7.
- 37. Dewis J, Freitas F. 1970. Physical and chemical methods of soil and water analysis. FAO
Soils Bulletin:275 pp.
- 38. Kamara A, Rhodes ER, Sawyerr PA. 2007. Dry Combustion Carbon, Walkley–Black
Carbon, and Loss on Ignition for Aggregate Size Fractions on a Toposequence.
*Communications in Soil Science and Plant Analysis* 38:2005-2012.
- 39. Bremner JM. 1965. Total Nitrogen, p 1149-1178, *Methods of Soil Analysis*
doi:<https://doi.org/10.2134/agronmonogr9.2.c32>.
- 40. Keeney DR, Nelson DW. 1982. Nitrogen—Inorganic Forms, p 643-698, *Methods of Soil*
*Analysis* doi:<https://doi.org/10.2134/agronmonogr9.2.2ed.c33>.
- 41. Parada AE, Needham DM, Fuhrman JA. 2016. Every base matters: assessing small
subunit rRNA primers for marine microbiomes with mock communities, time series and
global field samples. *Environ Microbiol* 18:1403-14.
- 42. Walters W, Hyde ER, Berg-Lyons D, Ackermann G, Humphrey G, Parada A, Gilbert JA,
Jansson JK, Caporaso JG, Fuhrman JA, Apprill A, Knight R. 2016. Improved Bacterial
16S rRNA Gene (V4 and V4-5) and Fungal Internal Transcribed Spacer Marker Gene
Primers for Microbial Community Surveys. *mSystems* 1.
- 43. Anonymous. 2023. R CORE Teame . A language and environment for statistical
computing. , R Foundation for Statistical Computing<https://www.R-project.org>, Vienna,
Austria.
- 44. Callahan BJ, McMurdie PJ, Rosen MJ, Han AW, Johnson AJ, Holmes SP. 2016. DADA2:
High-resolution sample inference from Illumina amplicon data. *Nat Methods* 13:581-3.

45. Lin H, Peddada SD. 2020. Analysis of microbial compositions: a review of normalization
and differential abundance analysis. NPJ Biofilms Microbiomes 6:60.

Re: Spectrum02963-24R1 (**The soil microorganism *Bacillus amyloliquefaciens* N3-8 shows potential as a biocontrol agent against the pathogen *Burkholderia pseudomallei* and its effect on rice plantation**)

Dear Dr. Rasana W. Sermswan:

The comments made by two independent reviewers have been attended satisfactorily and I am glad to be able to accept your manuscript for publication.

Your manuscript has been accepted, and I am forwarding it to the ASM production staff for publication. Your paper will first be checked to make sure all elements meet the technical requirements. ASM staff will contact you if anything needs to be revised before copyediting and production can begin. Otherwise, you will be notified when your proofs are ready to be viewed.

Sincerely,
Frédérique Reverchon
Editor
Microbiology Spectrum